# A method for solving heat transfer with phase change in ice or soil that allows for large time steps while guaranteeing energy conservation

Niccolò Tubini[1], Stephan Gruber[2], and Riccardo Rigon[1]

[1]Department of Civil, Environmental and Mechanical Engineering, Universtity of Trento, Trento, Italy
[2]Department of Geography and Environmental Studies, Carleton University, Ottawa, ON, K1S 5B6, Canada

**Correspondence:** Niccolò Tubini (niccolo.tubini@unitn.it)

**Abstract.** The accurate simulation of heat transfer with phase change is a central problem in cryosphere studies. This is because the nonlinear behaviour of enthalpy as function of temperature can prevent thermal models of snow, ice and frozen soil from converging to the correct solution. Existing numerical techniques rely on increased temporal resolution in trying to keep corresponding errors withing acceptable bounds. Here, we propose an algorithm, originally applied to solve water flow in soils, as a method to solve these integration issues with guaranteed convergence and conservation of energy for any time step size.

We review common modeling approaches, focusing on the fixed-grid method and on frozen soil. Based on this, we develop a conservative formulation of the governing equation and outline problems of alternative formulations in discretized form. Then, we apply the nested Newton-Casulli-Zanolli (NCZ) algorithm to a one-dimensional finite-volume discretization of the energy-enthalpy formulation.

Model performance is demonstrated against the Neumann and Lunardini analytical solutions and by comparing results from numerical experiments with integration time steps of one hour, one day, and ten days. Using our formulation and the NCZ algorithm, the convergence of the solver is guaranteed for any time step size. With this approach, the integration time step can be chosen to match the time scale of the processes investigated.

## 1   Introduction

Freezing and thawing of soils affect a wide range of biogeochemical and hydrological (Walvoord and Kurylyk, 2016; Schuur et al., 2015) processes and interact with engineered structures in cold regions. Correspondingly, the simulation of freezing and thawing soil is an important an well-researched topic (Streletskiy et al., 2019; Harris et al., 2009). Climate change brings additional urgency and new phenomena of interest to these studies. It is thus not a surprise that many models of freezing and

thawing soil and ice exist, some of which are reviewed in Appendix A. Here, we propose a solution to a central challenge that these models have in common.

Published models can be categorized as empirical, analytical, or numerical (Riseborough et al., 2008). Empirical methods relate ground temperature or thawing/freezing depth (TFD) to simple topoclimatic factors (Zhang et al., 2008; Riseborough et al., 2008) and are relatively simple to apply. By contrast, analytical and numerical models are based on the conservation of mass and energy and can be divided in two broad groups (Tan et al., 2011). The first group focuses primarily on freezing and thawing, commonly known as the Stefan problem. The governing equation describes energy conservation with the heat flux modelled using the Fourier law. The second group considers the coupled problem of heat transfer and water flow in soils. In this case energy-entalphy conservation equation includes also the advective heat flux and it is coupled with the mass conservation equation. For both groups, the latent heat transfer during phase change of water leads to problems related to convergence, conservation, and restrictions to discretization of space and time (Bao et al., 2016).

Historically (Hu and Argyropoulos, 1996; Vuik, 1993), the first attempts to solve the problem of heat conduction considering the phenomena of solidification and melting date back to the studies by Lame and Clapeyron in 1831, and the analytical solutions presented by Stefan around 1890, and Neumann in 1921. Later, other analytical solutions were proposed in order to overcome some simplifications that were too restrictive (Zhang et al., 2008; Riseborough et al., 2008; Walvoord and Kurylyk, 2016). These analytical solutions, however, are limited to one dimensional problems and constrained in their initial and boundary conditions as well as the description of soil characteristics (Kurylyk et al., 2014a).

By contrast, numerical models can accommodate complex processes or configurations, including soil heterogeneities, complicated temperature boundary conditions, intermittent freeze-thaw and temporally variable thermal properties. Accurately representing phase transitions, however, is a non-trivial task and several different methods have been published. They can be broadly cast in two general groups: the so-called front-tracking methods and the fixed-grid methods (Voller et al., 1990). Even though this contribution is focused on modelling heat transfer in frozen soil or ice, the following review includes, and is relevant for, other fields of research that involve phase change.

Front-tracking methods are suitable whenever the two phases are divided by a spatially smooth and continuous front and thus the state of the system can be conveniently described by the position of this interface (Voller et al., 1990). The moving front is tracked defining a continuity ('Stefan') condition on the heat flux across it. For example, the one-dimensional model by Goodrich (1978; 1982) uses front-tracking in modelling frozen soil and the SICOPOLIS model (Greve, 1997a, b; Greve and Blatter, 2016) uses it to model polythermal ice sheets.

In frozen soil, however, a significant proportion of water can remain liquid at temperatures well below 0 °C. This depression of the melting temperature is due to the presence of solutes (Bouyoucos, 1913; Bouyoucos and McCool, 1915; Bouyoucos, 1920, 1923), surface effects in the interaction between water and soil particles as well as water and ice (Anderson and Tice, 1972; Clow, 2018), and the Gibbs-Thomson effect (Rempel et al., 2004; Watanabe and Mizoguchi, 2002). To some degree, also polycrystalline ice has a temperature-dependent liquid water content (Langham, 1974). The gradual phase change over a range of temperatures in soils is commonly described with the soil freezing characteristic curve (SFCC) (Kurylyk and Watanabe,

2013). Moreover the presence of a partially frozen region is also common in ice and snow where liquid and solid phase coexist in thick isotherm layers.

With phase change occurring over a range of temperatures, rather than at one specific temperature, front-tracking methods become computationally expensive (Voller et al., 1990) and conceptually ambiguous. This is the case in many industrial (Voller and Cross, 1981) and environmental problems. Additionally, front tracking is complicated because it requires either a deforming grid or a transformed coordinate system (Aschwanden and Blatter, 2009). By contrast, fixed-grid methods can accurately describing the thermodynamics of the problem without requiring additional complications in handling the computational domain. For these reasons, fixed-grid methods are generally preferable to front-tracking methods when simulating frozen soil.

Fixed-grid methods include the latent heat of fusion in their governing equation, avoiding the necessity to define a continuity condition across the moving boundary and related implementation problems. All contemporary fixed-grid methods we reviewed aim to solve the numerical integration using globally convergent algorithms. Three differing approaches for treating the latent heat of fusion exists: the enthalpy method, using a source term, and using apparent heat capacity. As analytical expressions, these methods look the same because their governing equations can be obtained from each other by the chain rule of derivation. As we will illustrate in the next section, problems can arise in the discrete domain where this rule is not always valid.

Here we present a numerical model of heat conduction with freezing and thawing in soils without water flow that guarantees exact energy conservation for any time step size and for a wide range of soil freezing characteristics. It is novel in using the nested Newton-Casulli-Zanolli (NCZ) algorithm (Casulli and Zanolli, 2010) for solving the nonlinear system obtained from discretizing the governing equation, written in terms of the specific enthalpy, using a semi-implicit finite volume scheme. The NCZ algorithm has previously been applied to solving water flow in soils and to our knowledge this is first application for solving the heat equation. Long time steps are desirable in several applications including permafrost thaw, or surface components of climate models, and models dedicated to avalanche prediction.

The remainder of the paper is organized as follows. Section 2 reviews established approaches to study freezing and thawing phenomena in soils and points to relevant issues. Section 3 describes the new approach we propose. It details the discretization of the governing equation and the NCZ algorithm used to solve the resulting nonlinear numerical system. In Section 4, the new model is tested against analytical solutions and in Section 5, its performance is compared over a range of spatial and temporal resolutions. Section 6 summarises our findings and concludes this contribution. The Appendix B contains pseudocode to facilitate the implementation of the method we describe in other models.

## 2  The governing equation and their numerical issues

The governing equation of the problem in the first of the three approaches is written in terms of both the total enthalpy and temperature

$$\frac{\partial h(T)}{\partial t} = \nabla \cdot [\lambda(T)\nabla T] \tag{1}$$

where $h(T)$ is the specific enthalpy, $T$ is temperature, $\lambda(T)$ is the thermal conductivity, and $t$ is the time.

In the approach relying on apparent heat capacity, the governing equation is

$$C_a \frac{\partial T}{\partial t} = \nabla \cdot [\lambda(T)\nabla T] \qquad (2)$$

where

$\quad C_a = \dfrac{\partial h}{\partial T} = C_T + l_f \dfrac{\partial \theta_w}{\partial T} \qquad (3)$

is the apparent heat capacity that is the sum of the actual heat capacity $C_T$ and a term representing the additional thermal capacity arising from phase change with the local derivative of the SFCC (Dall'Amico, 2010).

In the approach using a source term for latent heat, it is considered as a heat source

$$C_T \frac{\partial T}{\partial t} = \nabla \cdot [\lambda(T)\nabla T] - l_f \frac{\partial \theta_w}{\partial T} \; , \qquad (4)$$

and in this equation, there are two unknowns: the temperature, and the liquid fraction $\theta_w$ appearing in the source term.

The specific enthalpy per unit mass is defined as

$$h = u + pv \qquad (5)$$

where $u$ is the specific internal energy, $p$ is pressure, and $v$ is the specific volume, the inverse of density. Assuming that the heat transfer occurs at constant pressure and volume the differential of the specific energy and of the specific enthalpy are equal

(Appendix D). However, since the term enthalpy method is commonly used in the literature, we will refer to enthalpy instead of internal energy.

The specific enthalpy of a control volume of soil $V_c$ can be calculated as the sum of the enthalpy of the soil particles, liquid water and ice (Dall'Amico et al., 2011):

$$h = h_{sp} + h_w + h_i \qquad (6)$$

Defining a reference temperature $T_{ref}$ the above terms becomes

$$h_{sp} = \rho_{sp} c_{sp} (1 - \theta_s)(T - T_{ref})$$
$$h_w = \rho_w c_w \theta_w(T)(T - T_{ref}) + \rho_w l_f \theta_w(T)$$
$$h_i = \rho_i c_i \theta_i(T)(T - T_{ref}) \qquad (7)$$

where $l_f$ is the specific latent heat of fusion, $\rho_{sp}$, $\rho_w$ and $\rho_i$ are the densities of the soil particles, water, and ice, $c_{sp}$, $c_w$, $c_i$ are the specif heat capacity of the soil particles, water, and ice, $\theta_w(T)$ is the unfrozen water content, and $\theta_i(T)$ is the ice content.

The liquid water content and the ice content are evaluated using SFCCs (Dall'Amico et al., 2011) which are dependent on temperature and, in the general case, on temperature and water saturation. Usually the reference temperature, $T_{ref}$, is set to 273.15 K, the melting temperature of pure water at standard atmospheric pressure. By using Eq. (7) the enthalpy Eq. (6) can be rewritten as

$$h = C_T(T - T_{ref}) + \rho_w l_f \theta_w(T) \qquad (8)$$

where $C_T = \rho_{sp}c_{sp}(1 - \theta_s) + \rho_w c_w \theta_w(T) + \rho_i c_i \theta_i(T)$ is the bulk heat capacity of the soil volume $V_c$.

SFCCs have an inflection point (Bao et al., 2016; Hansson et al., 2004) causing a sharp change in their derivative. This nonlinear behaviour gives rise to convergence problems during the solution of the system of equations resulting from the numerical approximation of the governing equation (Voller, 1990; Casulli and Zanolli, 2010). This is true for any method used such as finite differences (Westermann et al., 2016; Bao et al., 2016; Sergueev et al., 2003), finite elements (McKenzie et al., 2007), and finite volumes (Dall'Amico et al., 2011). As a consequence, the robustness (stability) of the numerics used is a fundamental and important issue in frozen soil models.

There is a more subtle aspect in the integration though. Analytically, Eq. (1, 2, and 4) are equivalent because Eq. (2 and 4) are derived from Eq. (1) by applying the chain rule of derivative on the enthalpy under the general assumption that the enthalpy is a differentiable variable. However, this is not necessarily so in the discrete domain where the derivative chain rule is not always valid. This is a known issue when dealing with hyperbolic equations (Roe, 1981), but often overlooked when treating the parabolic ones.

The apparent heat-capacity approach, Eq. (2) can suffer from large balance errors in the presence of high nonlinearities and strong gradients (Casulli and Zanolli, 2010). The key to deriving a conservative numerical method here concerns the discretization of the apparent heat capacity, and Nicolsky et al. (2007b) as well as Voller et al. (1990) discussed suitable techniques.

Referring to the work by Roe (Roe, 1981), it can be proven that, the discrete operator of $C_a$ has to satisfy the requirement

$$\tilde{C}_{a_i}^{n+1/2}(T_i^{n+1} - T_i^n) = h(T_i^{n+1}) - h(T_i^n) \,, \tag{9}$$

ensuring preservation of the chain rule at the discrete level. In Eq. (9) $i$ refers to the spatial discretization index and $n$ to the time discretization index. Approximating the time derivative in Eq. (2) using a backward Euler scheme we obtain

$$\tilde{C}_{a_i}^{n+1/2}\frac{T_i^{n+1} - T_i^n}{\Delta t} \,, \tag{10}$$

and substituting the condition Eq. (9)

$$\frac{h(T_i^{n+1}) - h(T_i^n)}{T_i^{n+1} - T_i^n}\frac{T_i^{n+1} - T_i^n}{\Delta t} = \frac{h_i^{n+1} - h_i^n}{\Delta t} \tag{11}$$

This shows that solving Eq. (2) with a conservative numerical method, i.e. by making use of Eq. (9), is equivalent to solving the enthalpy formulation, Eq. (1). Roe's condition, however, is often not checked in numerical models.

The source-term approach presents problems analogous to those of the apparent heat-capacity formulation. Specifically, Eq. (4) is derived from Eq. (2) by moving the latent heat term to the right-hand-side of the equation. Equation (4) can be solved numerically using an iterative procedure (Voller et al., 1990) or the Decoupled Energy Conservation Parametrization method (DECP) (Zhang et al., 2008). As pointed out by Voller et al. 1990, the numerical solutions based on an iterative procedure may suffer from non-convergence problem unless under-relaxation is wisely applied, and additionally, it necessary to guarantee that the liquid fraction is in the range $(0, 1)$. With DECP, the energy equation is first solved without latent heat. Then, soil temperature and the liquid and solid fractions are readjusted to ensure energy conservation during phase change.

This method is mainly used in land-surface models (LSMs) (Dai et al., 2003; Foley et al., 1996; Verseghy, 1991). In this case, Nicolsky et al. (2007a) showed that it results in an artificial stretch of the phase change region, with consequent inaccuracies in the simulation of active-layer thickness. A summary of relevant models is given in Table 1 and more details in Appendix A.

In summary, the governing equation can be written using three different approaches that are equivalent analytically, but not in their discrete formulation. Of the three, the enthalpy approach remains conservative, even when discretized, and should be preferred. An additional fundamental problem is the solution of the nonlinear system of equations. Current algorithms either require time step adaptation or may fail to converge, leading to unstable simulations and reduced computational efficiency (Casulli and Zanolli, 2010). Here we address this fundamental challenge by using the NCZ algorithm to solve the nonlinear

system of equations. Compared to other algorithms, it guarantees convergence of the solution for any integration time step. When the time step is not constrained by numerical issues, it can be chosen to better match the time scale of the process under investigation.

## 3    A soil heat-transfer model using the NCZ algorithm

Frozen soil models are typically solved with time steps between seconds and hours. This may be motivated by the desire to

resolve diurnal phenomena near the ground surface, and also, this often arises from limitations of the numerical schemes used. Many applications related to permafrost (Erum et al., 2019), on the other hand, only require the representation of seasonal and multi-annual variation, which can be accomplished using time steps of one or more days if permitted by the numerical schemes employed. In order to have a numerical scheme that does not suffer from time step restriction due to a stability condition, a semi-implicit time integration is required. A semi-implicit formulation includes the necessity of solving a nonlinear system

of equations and the algorithm used for this is of great importance. Existing linearization algorithms such as the Picard or the Newton, require a sufficiently accurate initial guess. As reported by Casulli and Zanolli (2010) this can be obtained by using small time steps, often requiring an empirical criterion for time-step adaptation. Therefore, even if the numerical scheme does not require a time step restriction, one may still be required to solve the nonlinear system of equations. Moreover, to the knowledge of the authors and colleagues (F. Gugole, M. Dumbser, G. Stelling, personal communication, 2019), the currently

used algorithms to solve nonlinear system do not offer a mathematically guaranteed convergence. This is important, because an inexact solution of the nonlinear system is not conservative (Casulli and Zanolli, 2010).

### 3.1    Discretization of the domain

The domain is partitioned using an unstructured orthogonal grid (Casulli and Walters, 2000), consisting of a set of nonoverlapping convex volumes $\Omega_i$, $i = 1, 2, .., N_v$, separated by $M$ internal faces $\Gamma_j$, $j = 1, 2, ..., M$. Let $\mathcal{A}_j$ denote the nonzero $j$th face

area. Within each control volume a centre must be identified in such a way that the segment joining the centres of two adjacent volumes and the face shared by the two volumes have a nonempty intersection, are orthogonal to each other and have distance $\delta_j$. Each control volume $\Omega_i$ may have an arbitrary number of faces. Let $\mathcal{F}_i$ denote the nonempty set of faces of the $i$th volume, with the exclusion of boundary faces. Moreover, let $\mathcal{P}(i, j)$ be the neighbour of volume $i$ that shares face $j$ with the $i$th control

**Table 1.** Summarizing table of relevant existing freezing models. More details are in Appendix A.

| Model | Form | Time discretization | Nonlinear solver | Limitation |
|---|---|---|---|---|
| CLM | N. L. H[a] | Crank-Nicolson | DECP[f] | Artificial stretch of phase change region[h] and non-convergence problem[i]. Monotonicity time step restriction. |
| CoupModel | A. H. C[b]. | Explicit | Not required | Stability time step restriction. |
| CryoGrid | A. H. C[b]. | Implicit | Newton based algorithm | Convergence is not guaranteed[g]. |
| GEOtop | A. H. C[b]. | Implicit | Globally convergent Newton algorithm | Convergence is not guaranteed[g]. |
| GIPL-2.0 | E. F.[c] | Implicit | Newton algorithm with Godunov splitting | Convergence is not guaranteed[g]. |
| Goodrich | N. L. H.[a] | Implicit | Front tacking method | Computationally expensive. Problems arise when the phase change occurs over a range of temperatures. |
| Hydrus 1D | A. H. C[b]. | Implicit | Picard iteration | Convergence is not guaranteed[g]. |
| MarsFlow | E. F.[c] | Implicit | Newton-Raphson algorithm | Convergence is not guaranteed[g]. |
| NEST | S. T.[d] | Explicit | Not required | Stability time step restriction. |
| SoilVision | A. H. C[b]. | Explicit and implicit | Newton-Raphson algorithm | Convergence is not guaranteed[g]. |
| SUTRA | A. H. C[b]. | Implicit | Picard iteration | Convergence is not guaranteed[g]. |
| Crocus | E. F.[c] | Implicit | DECP[f] | Artificial stretch of phase change region due to the DECP[h] and non-convergent problem[i]. |
| SNOWPACK | S. T | Implicit | DECP[f] | Artificial stretch of phase change region[h] and non-convergence problem[i]. |
| ORCHIDEE | N. L. H[a] | Explicit | DECP[f] | Artificial stretch of phase change region[h] and non-convergence problem[i]. |
| JSBACH | S. T.[d] | Implicit | DECP[f] | Artificial stretch of phase change region[h] and non-convergence problem[i]. |
| Aschwanden Blatter | E. G. M.[c] | Implicit | Newton based algorithm | Convergence is not guaranteed[g]. |
| SICOPOLIS | N. L. H.[a] | Implicit | Front tracking method with a transformed coordinate system | Computationally expensive. |
| Schoof Hewitt | E. F.[e] | Implicit and explicit | Not required | Requires the partition of the domain in cold and temperate regions. |

[a]The governing equation is written in only in terms of temperature and the latent heat is not included. [b]Apparent heat capacity formulation. [c]Enthalpy formulation. [d]Source term formulation. [e]The heat flux is written in terms of enthalpy and not of temperature as in the enthalpy formulation.

[f] Decoupled Energy Conservation Parametrization. [g]There are no guarantees that the nonlinear solver converge (Casulli and Zanolli, 2010, 2012).

[h] (Nicolsky et al., 2007b). [i] (Voller et al., 1990).

volume so that $1 \leq \mathcal{P}(i,j) \leq N_v$ for all $j \in \mathcal{F}_i$. The discrete variables $h_i$ and $T_i$ are located at centre of each element $\Omega_i$. Using a semi-implicit finite volume method, the discretization of Eq. (1) reads as

$$h_i(T_i^{n+1}) = h_i(T_i^n) + \Delta t \left[ \sum_{j \in \mathcal{F}_i} \Lambda_j^n \frac{T_{\mathcal{P}(i,j)}^{n+1} - T_i^{n+1}}{\delta_j} + S_i^n \right] \tag{12}$$

where $\Delta t$ is the time step size,

$$\Lambda_j^n := \mathcal{A}_j \max \left[ \lambda_i(T_i^n), \lambda_{\mathcal{P}(i,j)}(T_{\mathcal{P}(i,j)}^n) \right] \tag{13}$$

and

$$S_i = \int_{\Omega_i} S \, d\Omega \tag{14}$$

is an optional source/sink term in volume, and $h_i(T)$ is the $i$th enthalpy given by

$$h_i(T) = \int_{\Omega_i} h(T) \, d\Omega. \tag{15}$$

Eq. (12) can be written in matrix form as

$$\boldsymbol{h}(\boldsymbol{T}) + \mathbf{A} \boldsymbol{T} = \boldsymbol{b} \tag{16}$$

where $\boldsymbol{T} = \{T_i\}$ is the vector of unknowns, $\boldsymbol{h}(\boldsymbol{T}) = h_i(T_i)$ is a vectorial function representing the discrete enthalpy, $\mathbf{A}$ is the energy flux matrix, and $\boldsymbol{b}$ is the right-hand-side vector of Eq. (12), which is properly augmented by the known Dirichlet boundary condition when necessary. For a given initial condition $T_i^0$, at any time step $n = 1, 2, \dots$ Eq. (12) constitutes a nonlinear system for $T_i^{n+1}$, with the nonlinearity affecting only the diagonal of the system and being represented by the enthalpy $h_i(T_i^{n+1})$. This set of equations is a consistent and conservative discretization of Eq. (1). Therefore, regardless of the chosen spatial and temporal resolution, $T_i^{n+1}$ is a conservative approximation of the new temperature.

### 3.2 Solution of the nonlinear system

Difficulties in solving the nonlinear system of Eq. (16) arise from the non-monotonic behaviour of the derivative of the enthalpy, $h(T)$, with respect to temperature, and because in some parametrizations used for substances such as water, the derivative of the enthalpy is not correctly defined. The NCZ algorithm used here was discovered by Casulli and Zanolli (2010) and overcomes these difficulties with a nested Newton algorithm, two subsequent Newton-type iterations. NCZ is based on Jordan decomposition (Chistyakov, 1997) of the enthalpy function, rewriting it as the difference of two monotonic functions on which the Newton algorithm can be applied separately in a nested iteration, as explained below. A mathematical proof of convergence exists for NCZ (Brugnano and Casulli, 2008, 2009; Casulli and Zanolli, 2010, 2012).

For each control volume the enthalpy function $h_i(T)$ can be defined as

$$h_i(T) = \int_{T_{ref}}^{T} C_{a,i}(\xi) d\xi \tag{17}$$

where $C_{a,i}(T)$ is is defined as $C_{a,i}(T) = \int_{\Omega_i} C_{a,i}(T)d\Omega$. $C_{a,i}(T)$ has to fulfil two requirements. The first one, C1, is that $C_{a,i}(T)$ is defined for every $T$ and that it is a nonnegative function function with bounded variations. The second one, C2, is that there exist a $T^*$ such that $C_{a,i}(T)$ is strictly positive and non decreasing in $(0, T^*)$ and nonincreasing in $(T^*, +\infty)$.

Since $C_{a,i}(T)$ are nonnegative functions with bounded variations, they are almost everywhere differentiable, admit only discontinuities of the first kind, and can be expressed by using the Jordan decomposition (Fig.1) as the difference of two nonnegative, nondecreasing, and bounded functions:

$$p_i(T) = C_{a,i}(T) \qquad\qquad q_i(T) = 0 \qquad\qquad\qquad if\, T \leq T^* \qquad (18)$$
$$p_i(T) = C_{a,i}(T^*) \qquad\qquad q_i(T) = p_i(T) - C_{a,i}(T) \qquad\qquad if\, T > T^*$$

Using Eq. (18) the enthalpy $h_i(T)$ can be written as $h_i(T) = h_{1,i}(T) - h_{2,i}(T)$ where:

$$h_{1,i}(T) = h_i(T) \qquad\qquad\qquad\qquad\qquad\qquad\qquad if\, T \leq T^*$$
$$h_{1,i}(T) = h_i(T^*) + C_{a,i}(T^*)(T - T^*) \qquad\qquad\qquad if\, T > T^*$$

$$(19)$$

$$h_{2,i}(T) = 0 \qquad\qquad\qquad\qquad\qquad\qquad\qquad\qquad if\, T \leq T^*$$
$$h_{2,i}(T) = h_{1,i}(T) - h_i(T) \qquad\qquad\qquad\qquad\qquad if\, T > T^*$$

By making use of Eq. (19) the algebraic system, Eq. (16), can be written as

$$\boldsymbol{h_1}(\boldsymbol{T}) - \boldsymbol{h_2}(\boldsymbol{T}) + \mathbf{A}\boldsymbol{T} = \boldsymbol{b} \qquad (20)$$

The NCZ algorithm requires first the linearization of $\boldsymbol{h_2}$ and then the linearization of $\boldsymbol{h_1}$. By choosing the initial guess for the NCZ algorithm such that $\boldsymbol{T}^0 \leq \boldsymbol{T}^*$, a sequence of outer iterates $\{T^k\}$ is obtained from Eq. (20) by linearizing $\boldsymbol{h_2}$ as follows:

$$\boldsymbol{h_1}(\boldsymbol{T}^k) - [\boldsymbol{h_2}(\boldsymbol{T}^{k-1}) + \boldsymbol{Q}(\boldsymbol{T}^{k-1})(\boldsymbol{T}^k - \boldsymbol{T}^{k-1})] + \mathbf{A}\boldsymbol{T}^k = \boldsymbol{b} \qquad (21)$$

so that the outer iterates are solutions of the following nonlinear system:

$$\boldsymbol{h_1}(\boldsymbol{T}^k) + (\mathbf{A} + \boldsymbol{Q}^{k-1})\boldsymbol{T}^k = \boldsymbol{d}^{k-1}, \quad k = 1, 2, \dots \qquad (22)$$

where $\boldsymbol{Q}^{k-1} = \boldsymbol{Q}(\boldsymbol{T}^{k-1})$ and $\boldsymbol{d}^{k-1} = \boldsymbol{b} + \boldsymbol{h_2}(\boldsymbol{T}^{k-1}) - \boldsymbol{Q}^{k-1}\boldsymbol{T}^{k-1}$. The resulting $k$th (outer) residual is derived form Eq. 20 and reads as follow

$$\boldsymbol{r}^k = \boldsymbol{h_1}(\boldsymbol{T}^k) - \boldsymbol{h_2}(\boldsymbol{T}^k) + \mathbf{A}\boldsymbol{T}^k - \boldsymbol{b} \qquad (23)$$

For all $k = 1, 2, \dots$, by setting $\boldsymbol{T}^{k,0} = \boldsymbol{T}^{k-1}$, a sequence of inner iterates $\{\boldsymbol{T}^{k,l}\}$ is derived from Eq. (22) by linearizing $\boldsymbol{h_1}(\boldsymbol{T})$ as follows

$$[\boldsymbol{h_1}(\boldsymbol{T}^{k,l-1}) + \boldsymbol{P}(\boldsymbol{T}^{k,l-1})(\boldsymbol{T}^{k,l} - \boldsymbol{T}^{k,l-1})] + (\mathbf{A} + \boldsymbol{Q}^{k-1})\boldsymbol{T}^{k,l} = \boldsymbol{d}^{k-1}, \quad l = 1, 2, \dots \qquad (24)$$

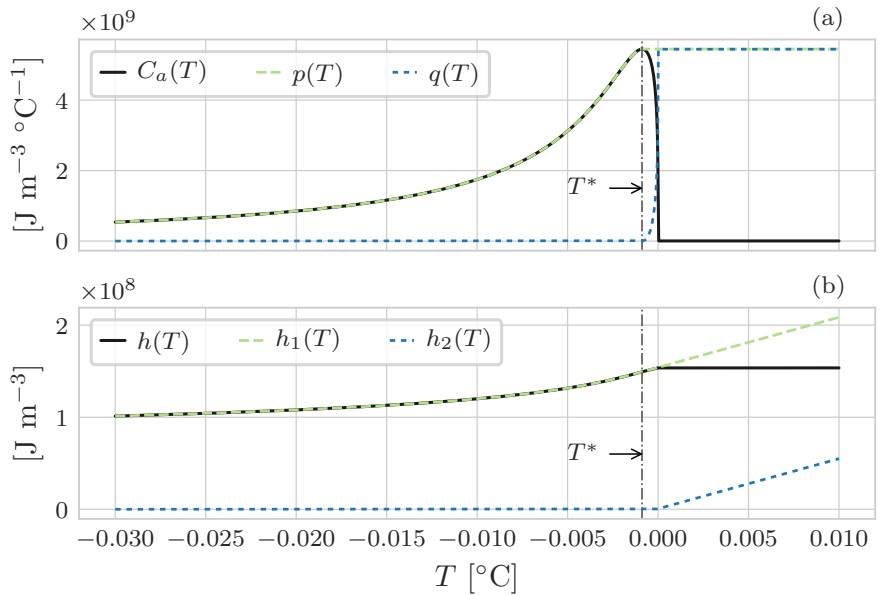

**Figure 1.** Graphical representation of the Jordan decomposition for the enthalpy of soil using the SFCC model for a silty soil (Dall'Amico, 2010). (a) shows the Jordan decomposition of $C_a(T)$, Eq. (18). For $T = T^*$, $C_a(T)$ presents a maximum: for $T < T^*$ it is increasing, and for $T > T^*$ it is decreasing. This non monothonic behaviour causes problems when solving the nonlinear system. $C_a(T)$ is thus replaced by $p(T)$ and $q(T)$, two monotonic functions. Consequently, (b), $h(T)$ is replaced by $h_1(T)$ and $h_2(T)$, Eq. (19)

so that the inner iterates are determined from the following linear systems

$$(\boldsymbol{P}^{k,l-1} + \mathbf{A} + \boldsymbol{Q}^{k-1})\boldsymbol{T}^{k,l} = \boldsymbol{f}^{k,l-1}, \quad l = 1, 2, \ldots \tag{25}$$

where $\boldsymbol{P}^{k,l-1} = \boldsymbol{P}(\boldsymbol{T}^{k,l-1})$ and $\boldsymbol{f}^{k,l-1} = \boldsymbol{d}^{k-1} - \boldsymbol{h_1}(\boldsymbol{T}^{k,l-1}) + \boldsymbol{P}^{k,l-1}\boldsymbol{T}^{k,l-1}$. The resulting $(k, l)$th (inner) residual is derived form Eq. 22 and reads as follow

$$\boldsymbol{r}^{k,l} = \boldsymbol{h_1}(\boldsymbol{T}^{k,l}) + (\mathbf{A} + \boldsymbol{Q}^{k-1})\boldsymbol{T}^{k,l} - \boldsymbol{d}^{k-1}. \tag{26}$$

The inner and the outer iterations are terminated when $\|\boldsymbol{r}^{k,l}\| < \varepsilon$, and $\|\boldsymbol{r}^k\| < \varepsilon$, respectively, with $\varepsilon$ being a sufficiently small prefixed tolerance.

The most commonly used constitutive SFCCs (McKenzie et al., 2007; Kozlowski, 2007; Dall'Amico et al., 2011; Sheshukov and Nieber, 2011; Watanabe et al., 2011), used to define the enthalpy of frozen soil, satisfy the assumptions C1 and C2. In particular, the NCZ approach can be successfully applied to SFCCs models derived from the combination of existing SWRC models and the Clapeyron equation that in general are difficult to implement in numerical models based on the apparent heat capacity (Kurylyk and Watanabe, 2013). Functions describing the internal energy of other substances, for instance pure water

(Andreas et al., 2005), satisfy the assumptions C1 and C2 and, therefore, the NCZ method can be successfully used to model phase change problems as it will be shown for the original Stefan problem, Sections (4.1.

**Table 2.** Maximum error m of the freezing front position from the numerical solution with the NCZ algorithm for different space and time discretizations relative to the Neumann analytical solution. For the numerical solution the position of the freezing front has been reconstructed from the linear interpolation of the temperature profile.

|                       | $\Delta t = 60$ s | $\Delta t = 300$ s | $\Delta t = 3600$ s |
|-----------------------|-------------------|--------------------|---------------------|
| $\Delta z = 0.001$ m  | 0.00737           | 0.00153            | 0.00739             |
| $\Delta z = 0.005$ m  | 0.00271           | 0.00302            | 0.00714             |
| $\Delta z = 0.01$ m   | 0.00536           | 0.00553            | 0.00905             |

## 4 Analytical Benchmarks

The numerical model is compared for the problem of a column of freezing water, i.e. the Stefan problem, with the analytical solution presented by Neumann (cited in Kurylyk et al. (2014b)), and for the problem of a column of soil with the three-zone with the analytical solution presented by Lunardini (1988).

### 4.1 Neumann analytical solution

The Neumann analytical solution gives the solution of unilateral freezing of a semi-infinite domain for both the temperature profile and the position of the moving boundary. Kurylyk et al. (2014b) recommended the Neumann solution due to its ability to represent differences between the thermal diffusivities of the thawed and frozen zones. Here we consider the freezing of pure water instead of soil since it is more numerically demanding. Consider a semi-infinite domain of pure water at temperature $T(z, 0) = T_0$ where $T_0 > T_m$, Fig. (2). At the surface a Dirichlet boundary condition is imposed $T(z = 0, t) = T_s$, with $T_s < T_m$. As a consequence a freezing front $\zeta$ propagates downward separating the solid and the liquid phase. The governing equations are

$$\frac{\partial h}{\partial t} = \lambda \frac{\partial^2 T}{\partial z^2} \tag{27}$$

$$T(\zeta, t) = T_m \tag{28}$$

$$\lambda_i \frac{\partial T}{\partial z}\bigg|_{z=\zeta^-} dt = \lambda_w \frac{\partial T}{\partial z}\bigg|_{z=\zeta^+} dt + l_f \rho \, d\zeta \tag{29}$$

At the moving boundary $\zeta(t)$, the temperature is equal to the melting temperature of water, and the time evolution of $\zeta(t)$ is described by the third equation, the Stefan condition. This condition states that the difference of the heat fluxes at the interface of the two substances is consumed for the phase change. The parameters used in the comparison are given in Table E1. The numerical model is able to simulate the freezing problem of water well as seen in Fig. (3) and Fig. (4).

For comparison, Kurylyk et al. 2014b tested the numerical model SUTRA against the Neumann analytical solution considering a soil porosity of $0.50 \, \mathrm{m^3 m^{-3}}$. For their test the time step was of $0.04 - 0.4$ s, the vertical spatial discretization 0.001 m,

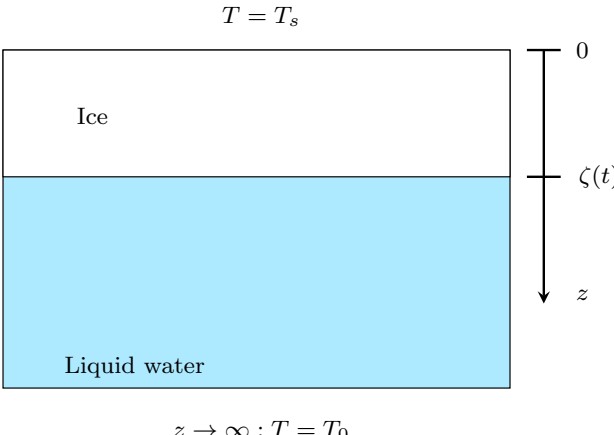

$T = T_s$

Ice

$0$

$\zeta(t)$

$z$

Liquid water

$z \to \infty : T = T_0$

**Figure 2.** Scheme showing the setting of the Neumann solution for the freezing case. Initially all water is liquid, $T_0 > T_m$. Because of the surface boundary condition, $T_s < T_m$, a freezing front, $\zeta$, propagates downward.

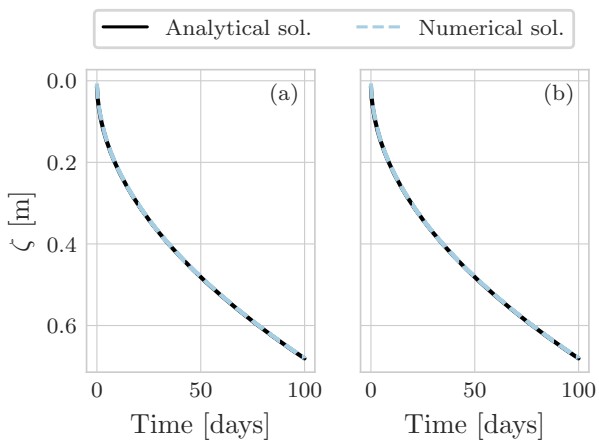

**Figure 3.** Propagation of the freezing front compared between the Neumann analytical and the numerical solution with the NCZ algorithm. Two space discretizations are used: (a) $\Delta z = 0.005$ m, and (b) $\Delta z = 0.001$ m. The integration time step is $\Delta t = 3600$ s.

and the parameter $\epsilon$ was increased to $-0.01$ °C to match the analytical solution. The maximum absolute error of the freezing front position was $0.00099$ m.

In our model, the choice of a small melting temperature range $\epsilon = 0.0001$ °C does not affect the quality of the numerical solution even at a large time step of 3600 s. Looking at Table 2 it is clear that the choice of the time step size is somehow related to the choice of the spatial discretization: using a small time step with a coarse grid does not necessarily improve the accuracy of the position of the freezing front.

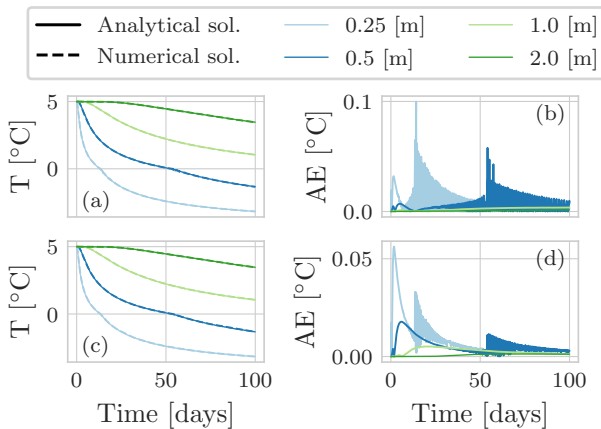

**Figure 4.** Panels (a) and (c) show the temperature evolution for the Neumann analytical and the numerical solution with the NCZ algorithm at various depths for a spatial discretization $\Delta z = 0.005$ m and $\Delta z = 0.001$ m respectively. The integration time step is $\Delta t = 3600$ s. Panels (b) and (d) show the absolute error.

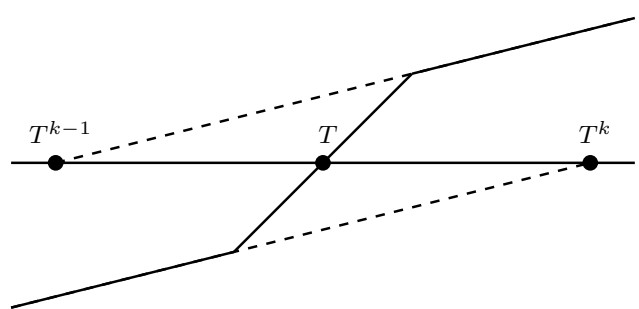

**Figure 5.** A scheme of problem which illustrates how the Newton-Raphson method can not converge towards T (Dall'Amico, 2010). In this case, the Newton-Raphson method fails to converge to $T$ since it cycles between $T^k$ and $T^{k+1}$ values.

We use the Neumann analytical solution to asses the the robustness of the NCZ algorithm in comparison with the Newton-Raphson and globally-convergent Newton methods. As reported by Dall'Amico et al. (2011), Figure (5) represents a well known case for which the Newton-Raphson algorithm can not converge. Instead, the solution continuously cycles between the iterates $T^{k-1}$ and $T^k$. While the Newton-Raphson algorithm converges to the exact solution if a good initial guess for $T^k$ exists, this represents a severe constraint for the reliable application for an iterative algorithm in a numerical model. An improvement of the Newton-Raphson algorithm can be obtained using the globally convergent Newton scheme (Dall'Amico et al., 2011). It uses the Newton-Raphson algorithm to provide the right search direction and, in order to avoid overshooting, a reduction factor $\delta$ is used to find the new estimate. This represents an improvement over the Newton-Raphson method, but its ability to converge depends on the choice of the parameter $\delta$ and on the treatment of the apparent heat capacity (Hansson et al., 2004; Nicolsky et al., 2007b; Dall'Amico et al., 2011). As such, this algorithm does not guarantees to converge for any time

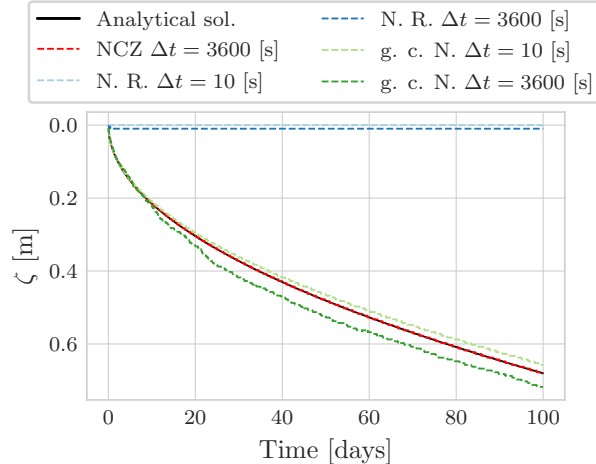

**Figure 6.** Comparison between the Neumann analytical solution and the numerical solution obtained with Newton-Raphson (N. R.), globally convergent Newton (g. c. N.), and NCZ algorithms. All the numerical simulations use the same spatial discretization $\Delta z = 0.005$ m.

step size and the requirements for small time steps can become a limiting factor. For example, in (Dall'Amico et al., 2011) the comparison between the Neumann solution and GEOtop has been done with a time step of $10$ s.

A comparison of the numerical solutions obtained with the Newton-Raphson algorithm, globally convergent Newton algorithm, and the NCZ algorithm shows significant differences (Fig. 6). Newton-Raphson cannot reproduce the analytical solution even if a time step of $\Delta t = 10$ s is used. The globally convergent Newton is in good agreement with the analytical solution if $\Delta t = 10$ s. With an hourly time step, however, the example with the globally convergent Newton method is not able to reproduce the position of the freezing front over longer periods of time. By contrast, the NCZ algorithm reproduces the analytical solution well using $\Delta t = 3600$ s. The quality of the solution obtained with the globally convergent Newton algorithm depends not only on the time step duration but also on the definition of the parameter $\delta$ (Fig. 7). The additional necessity for an arbitrarily chosen parameter in the globally convergent Newton algorithm further underscores the robustness of the NCZ algorithm, for which convergence only depends on the right definition of Eq. (18) and Eq. (19).

## 4.2 Lunardini analytical solution

Lunardini (1988) derived an analytical solution (Appendix F) for the temporal evolution of temperature during the freezing of a semi-infinite and initially unfrozen soil column. In contrast to the Neumann analytical solution, in the Lunardini analytical solution the domain is divided into three regions (Fig. 8) on the basis of temperature: unfrozen, $T < T_m$, partially frozen, $T_m < T < T_f$, and fully frozen, $T > T_f$. The domain is initially unfrozen with $T = T_0 = 4$ °C. At the left boundary condition a Dirichlet boundary condition is imposed with $T(x = 0, t) = T_s = -6$ °C, and the right boundary temperature is kept equal to the initial condition, $T(z \to \infty, t) = T_0$. Because the left boundary condition, $T_s < 0$ °C, a freezing front propagates from left to right.

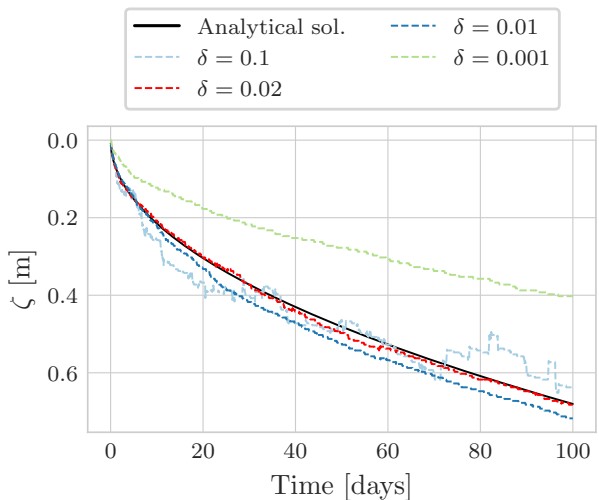

**Figure 7.** Comparison between the Neumann analytical solution and the numerical solution obtained with globally convergent Newton algorithm (g. c. N.). All the numerical simulations use the same spatial discretization $\Delta z = 0.005$ m and a time step size of $\Delta t = 3600$ s. This figure shows as the numerical solution depends on the choice of the parameter $\delta$.

**Table 3.** Maximum absolute error °C of the temperature after 24 h from the numerical solution with the NCZ algorithm relative to the Lunardini analytical solution. The space resolution is $\Delta x = 0.01$ m.

|                    | $T_m = -4\,°C$ | $T_m = -1\,°C$ | $T_m = -0.1\,°C$ |
|--------------------|----------------|----------------|------------------|
| $\Delta t = 300$ s  | 0.00683        | 0.01419        | 0.11436          |
| $\Delta t = 900$ s  | 0.01496        | 0.02448        | 0.11565          |
| $\Delta t = 3600$ s | 0.05115        | 0.08286        | 0.12116          |

We computed benchmark T1 proposed by the InterFrost project (InterFrost Project), parameters are given in Table (F1). The model agrees well with the analytical solution for all the three cases of $T_m$ in terms of both the temperature profile, Fig. (9) and Tab. (3), and the freezing front position, Fig. (10) and Tab. (4), even with an hourly time step.

For comparison, McKenzie et al. (2007) compared the numerical model SUTRA against the Lunardini analytical solution for the cases $T_m = -4\,°C$ and $T_m = -1\,°C$ using a time step size of $900$ s and a space resolution of $0.01$ m. For the first test case the maximum absolute error was $0.01\,°C$, and for the second $0.1\,°C$. Their parameters, however, differ from those suggested by the InterFrost consortium, making performance comparisons difficult. In particular, their porosity was $0.05$ m³m⁻³, whereas InterFrost uses $0.336$ m³m⁻³. As this determines the amount of latent heat involved in phase change, smaller errors are to be suspected to occur with the parameters used by McKenzie et al. (2007).

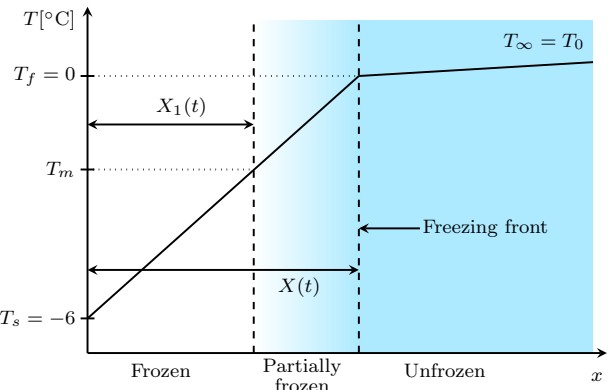

**Figure 8.** Scheme showing the setting of Lunardini problem (Ruhaak et al., 2015). Initially the domain is unfrozen with $T = T_0$. Because of $T_s < 0$ on the left boundary, a freezing front propagates from left to right. $X_1(t)$ and $X(t)$ identify respectively the isotherm corresponding to $T_m$, and $T_f$.

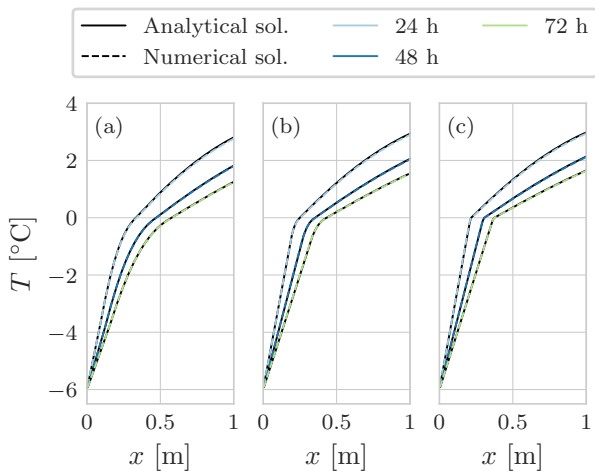

**Figure 9.** Comparison between the Lunardini solution and the numerical solution with the NCZ algorithm for the three cases of T1 benchmark: (a) $T_m = -4$ °C, (b) $T_m = -1$ °C, (c) $T_m = -0.1$ °C. The colours represent different times frame. The integration time step is $\Delta t = 3600$ s, and the space resolution is $\Delta x = 0.01$ m.

## 5 Numerical test

In the previous sections, we have demonstrated that the proposed method can reproduce the Neumann analytical solution, as well as the Lunardini analytical solution, even when using larger time steps than other numerical models.

After comparing simulation results with analytical solutions, we now analyse the difference between solutions using hourly, daily, and 10-day time steps. The domain is a soil column of 20 m depth that is uniformly at $T = -3$ °C, initially. The bottom

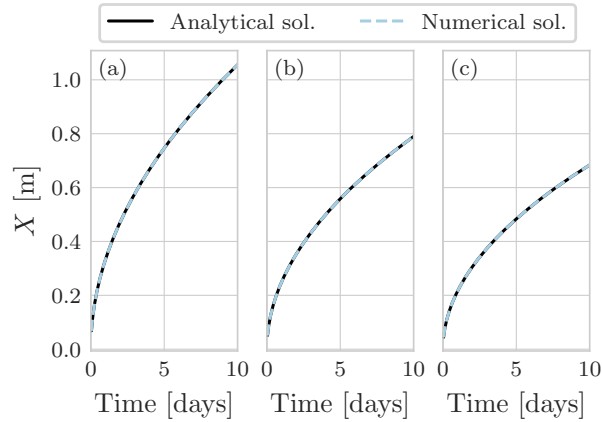

**Figure 10.** Propagation of the zero-isotherm for the Lunardini solution and the numerical solution with the NCZ algorithm for the three cases of T1 benchmark: (a) $T_m = -4\,°\text{C}$, (b) $T_m = -1\,°\text{C}$, (c) $T_m = -0.1\,°\text{C}$. The integration time step is $\Delta t = 3600$ s, and the space resolution is $\Delta x = 0.01$ m.

**Table 4.** Maximum error m of the freezing front position from the numerical solution with the NCZ algorithm relative to the Lunardini analytical solution. The space resolution is $\Delta x = 0.01$ m.

|  | $T_m = -4\,°\text{C}$ | $T_m = -1\,°\text{C}$ | $T_m = -0.1\,°\text{C}$ |
| --- | --- | --- | --- |
| $\Delta t = 300$ s | 0.00032 | 0.00051 | 0.00001 |
| $\Delta t = 900$ s | 0.00043 | 0.00027 | 0.00016 |
| $\Delta t = 3600$ s | 0.00062 | 0.00057 | 0.00047 |

boundary condition is adiabatic and at the surface, we use a Dirichlet boundary condition. The original forcing has hourly resolution and for longer time steps, corresponding averages are computed. As temperature gradients and the influence of phase change are usually greatest near the soil surface, the thickness $\Delta z$ is parameterized with an exponential function (Gubler et al., 2013)

$$\Delta z_i = \Delta z_{min}(1+b)^{i-1} \tag{30}$$

where $\Delta z_{min}$ is the thickness of the first layer, $b$ is the growth rate and $i$ is the layer index, being one at the ground surface and increasing downward. The parameters used are reported in Table G1. All three simulations were spun-up for a period of 1400 years to reach a stable thermal regime. After spin-up, we performed a simulation of 100 years.

Figure (11) compares the zero-isotherm position computed after 100 years for the three different time steps. Interestingly, there are no significant deviations in the results. The larger deviations occur when the zero-isotherm is shallow: at the beginning of the thawing season as well as the freezing one, Fig. (G1, G2). This can be attributed on one side to the diurnal cycles of

surface boundary condition, and on the other side that using a larger time step we lose accuracy in capturing the timing of thawing/freezing even if we use the same boundary condition.

With larger time steps, we lose some of the information of the boundary conditions and the accuracy of the numerical model decreases because it is first-order accurate in time. The overall performance relative to simulations with smaller time steps, however, is largely preserved. While the order of accuracy can be increased to second order in time using the Crank-Nicholson method, this would incur a time step restriction to guarantee the monotonicity of the solution. As this restriction is proportional to the square of the space discretization, $\Delta z^2$, the Crank-Nicholson method would represent a severe constrain whenever high
spatial resolution is required.

Figure (12) compares the minimum, mean, and maximum temperature profile respectively for the three simulations. (a) shows the ground temperature envelope for the hourly simulation. The maximum envelop presents an 'elbow' that is due to the phase-change effects Fig. (G3). As can be seen in (b) and (d), close to the soil surface the hourly simulation presents larger values for both the minimum and maximum temperature due the fact that the hourly boundary condition presents a greater
amplitude that is smoothed computing the daily and 10-day average.

In the mean temperature profile, the 10-day simulation presents a larger deviation from the hourly simulation than the daily simulation. The large deviation can be explained with the interaction of the time-step size with the thermal offset effect (Fig. G4). If the thermal conductivity of water is set equal to that of ice, the maximum difference between the three profiles is reduced to $0.003$ °C with a maximum deviation of $0.003$ °C from the initial condition, that is also equal to the mean of the
forcing boundary condition.

Regarding the spatial discretization Fig. (G5) reports a comparison of the zero-isotherm position obtained using an hourly time step, a daily time step, and a 10 day time step. The results are still in good agreement, but is it interesting to note that the zero-isotherm presents some steps, independently on the size of the time step, and some details are missed, such as the joining of the downward and upward freezing fronts captured with the finer grid. These steps are caused by the greater thickness of
the grid elements. Because temperature is computed in the middle of each control volume, more time is required to achieve complete phase change of water, resulting in slower variation of the zero-isotherm position.

These synthetic experiments demonstrate that spatial and temporal discretization can be chosen accordingly to the aim the study without any constrains due to the convergence and stability issues of the numerical scheme.

Moreover, we checked the mean number of iterations required to solve the nonlinear system for a simulation lasting 1
355    year with a time step $\Delta t = 1$ h and for different spatial discretizations. We compared the NCZ against the Newton-Raphson algorithm and the globally convergent Newton. Results are reported in Table G2.

## 6   Conclusions

We have presented a new model for simulating the ground thermal regime in the presence of freezing and thawing based on the heat-transfer equation and the application of the NCZ algorithm. To our knowledge, this is the only method that guarantees
convergence while also permitting large time steps. The numerical model was implemented and verified against the Neumann

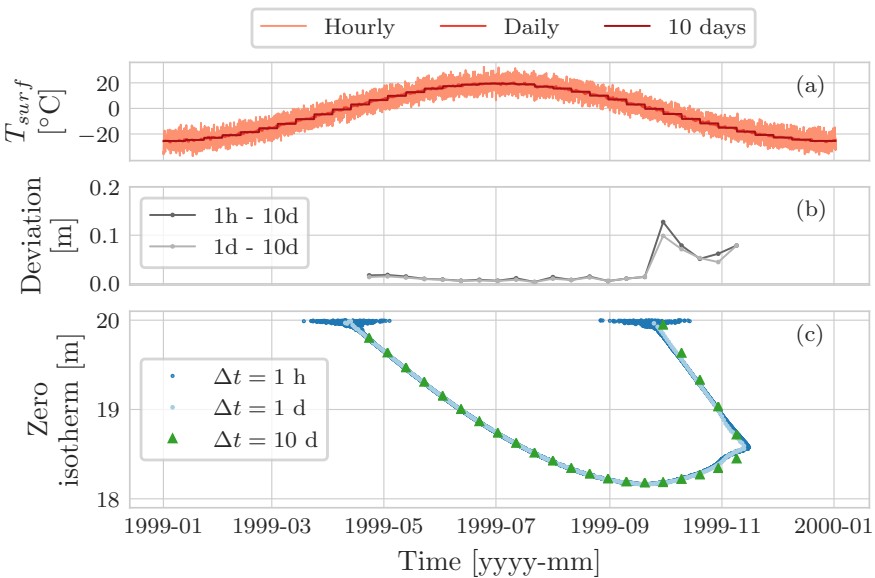

**Figure 11.** Comparison of the position of the zero-isotherm, panel (c), after 100 years of three simulations: using an hourly boundary condition with time step of $\Delta t = 1$ h, using a daily boundary condition with a time step of $\Delta t = 1$ day, and a 10-day boundary condition with a time step of $\Delta t = 10$ day. Panel (a) shows the surface temperature for the hourly, the daily and the 10-days simulations. Panel (b) shows the deviation of the position of the zero-isotherm after 100 years between the hourly and the 10-days simulation, and between the daily and the 10-days simulation.

and Lunardini analytical solutions. In both cases, the results were in good agreement even with an hourly integration time step. For the Neumann solution, we considered pure water instead of saturated soil since it is more numerically demanding, and no convergence problems were encountered despite choosing a narrow temperature range (0.0001 °C) over which phase change occurs.

Numerical experiments demonstrated the robustness of the model by comparing results at differing temporal and spatial resolutions. Results obtained with time steps of 1 h, 1 day, and 10 days are consistent. The robustness of the numerics allows the user to choose both the space and time discretization without any restriction due to stability and convergence issues. As a consequence, this method is effective for simulating permafrost thaw, a phenomenon that occurs at depth, in response to seasonal and multi-annual cycles, and often over tens, hundreds or even thousands of years. Furthermore, phenomena like

hysteresis or the variation of solute concentration upon freezing (Clow, 2018) can be included in the numerical model if the enthalpy function (i.e. its parameters) does not change within the current time step of integration.

     While we presented a finite volume method, the NCZ algorithm can be also used with finite difference and finite element method. Beyond applications to frozen soil, it can be used to study other geophysical phenomena that involve phase change of a substance simply by changing the definition of the enthalpy function and the thermal conductivity function. Examples include,

glacier dynamics (Aschwanden et al., 2012), snow pack evolution (Brun et al., 1992; Lehning et al., 1999), and magma bodies

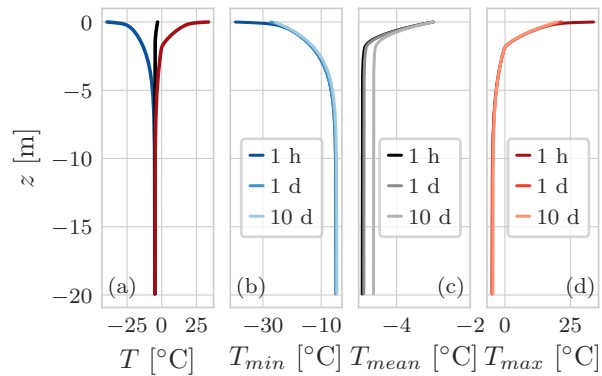

**Figure 12.** (a) The minimum, mean, and maximum temperature profile for the hourly simulation. (b), (c), (d) show the comparison of the minimum, mean, and maximum temperature profile respectively for the three simulations: with an hourly air temperature boundary condition and $\Delta t = 1$ h, with a daily air temperature boundary condition and $\Delta t = 1$ day, with a ten day air temperature boundary condition and $\Delta t = 10$ day. All three simulations last 100 years. The maximum difference of $T_{mean}$ between the hourly, and daily simulation is of 0.04 °C, while between the hourly, and ten-days simulation is of 0.3 °C.

(De Lorenzo et al., 2006). This may be even further expanded to industrial problems involving phase change materials used in energy recovery systems (Mongibello et al., 2018; Nazzi Ehms et al., 2019) or casting problems of pure metals and alloys (Lewis and Ravindran, 2000).

**codeavailability**

The source code is written in Java using the object-oriented programming paradigm. It can be found at https://github.com/ geoframecomponents/FreeThaw1D (Tubini, c). The OMS3 project can be found at https://github.com/GEOframeOMSProjects/ OMS_FreeThaw1D (Tubini, b). FreeThaw1D is deployed as an open source code to work alone or inside the Object Modelling System version 3 framework (David et al., 2013). In the latter case it can be connected at run time with the many other components developed along with the GEOframe system (Formetta et al., 2014; Bancheri, 2017) for providing hy-

drometeorological forcings and other fluxes, like the evapotranspiration. The simulations presented here can be found at http: //dx.doi.org/10.5281/zenodo.4017668 (Tubini, a). The code must be run inside the OMS3 Console or using the Dockerized version of OMS3. For setting up the environment please follow the steps described in the README file present it he Github repository https://github.com/GEOframeOMSProjects/OMS_FreeThaw1D and in the GEOframe pages at https://geoframe.blogspot. com/2020/12/installations-of-2021-geoframe.html (last access 28 January 2021). Once you have installed OMS3, please fol-

low the instructions contained in Jupyter_Notebook\_README.ipynb and Jupyter_Notebook\00_FreeThaw1D.ipynb. They contain all the details about the simulations inputs and parameters.

## Appendix A: Commonly used simulation software

The heat equation can be written in different forms that are analytically equivalent, but subject to differing numerical advantages and disadvantages. In the scientific literature, several simulators, i.e. software that implements a particular model (set of equations), for solving the heat equation with freezing and thawing have been presented. Here we review commonly used frozen soil models in terms of their governing equations and methods of finding numerical solutions.

Heat transfer with phase change of water is a cross-cutting problem existing in many geophysical phenomena other than frozen soil. This includes, for example, the seasonal snow pack, glaciers, and ice-sheets. Our contribution does not seek to present an improvement in the description of these problems and we ignore typical processes such as metamorphism and settling in seasonal snow or strain heating and deformation in glaciers and ice sheets. Nevertheless, corresponding models may benefit from the NCZ algorithm in the treatment of the nonlinearity arising from phase change and, furthermore, broadening our review to also include some snow and glacier models supports the generalisation of our findings.

### A1 CLM

The Community Land Model (CLM) is the LSM for the Community Earth System Model (Oleson et al., 2004). It includes a module to simulate the ground temperature considering freezing and thawing. The governing equation is written in the non-conservative form and does not include the latent heat term (Oleson et al., 2004) (Lawrence et al., 2019). The heat conduction equation is solved using a Crank-Nicholson method. The temperature profile is calculated adopting the DECP approach. This approach does not require to solve a nonlinear system, since the latent heat is treated in an explicit way, but Nicolsky et al. (2007a) have pointed out that this two-step procedure can overestimate the region where the phase change occurs, resulting in inaccuracies in the simulation of active-layer thickness.

### A2 CoupModel

The CoupModel (Jansson and Karlberg, 2011) is a one-dimensional numerical model to simulate the heat and water flow as well as carbon and nitrogen budgets in a soil-plant-atmosphere system (Hollesen et al., 2011). The governing equation for heat flow in the soil is defined using the apparent heat capacity, and solved with an explicit numerical method. This does not require to solve a non-linear system but sets a time step restriction to avoid numerical oscillation.

### A3 CryoGrid

CryoGrid 2 simulates the ground thermal regime based on conductive heat transfer in the soil and in the snowpack (Westermann et al., 2013). The heat equation is written using the apparent heat capacity and solved using the method of lines (Westermann et al., 2013). The resulting system of ordinary differential equations is solved numerically with the package CVODE of Sundials that implements a modified Newton method, and Inexact Newton method, or a fixed-point solver to linearize the algebraic system resulting from the discretization of the heat transfer equation. To our knowledge there is no mathematical proof that these algorithm converge to the exact solution.

## A4 GEOtop

GEOtop (Rigon et al., 2006; Endrizzi et al., 2014) is a physically based distributed model of the mass and energy balance of the
hydrological cycle. It includes a module for solving the energy equation in freezing soil (Dall'Amico et al., 2011); this module
can also be linked with the solver for the Richards equation. The governing equation for heat transfer is written in conservative
form, but when solving the equation the apparent heat capacity formulation is used. A globally convergent Newton algorithm
is used to deal with the non-linearities arising from phase change (Dall'Amico et al., 2011). The globally convergent Newton
represents an improvement over the Newton-Rapshon algorithm but it does not guarantee convergence of the solution, and as
presented in Section (4.1), the choice of the parameter $\delta$ is non trivial.

## A5 GIPL-2.0

GIPL-2.0 simulates the ground thermal regime by solving the heat equation with phase change numerically (Marchenko et al.,
2008). The governing equation is written in the conservative form and Newton's method is used to linearize the energy equation.
To overcome convergence problems when solving the non-linear system, GIPL-2.0 implements a fractional time step approach,
Godunov splitting. The key point of the solution regards the treatment of the enthalpy time derivative: in case of a non zero
gradient of temperature exists the time derivative is approximated with a difference derivative, otherwise using the analytical
representation.

## A6 Goodrich

Goodrich (1982) presented a one-dimensional model to simulate the ground thermal regime considering the phase change of
water. The governing equation is written in the non conservative form and does not include the latent heat of fusion. Phase
change is treated with the front tracking method, which offers good accuracy for problems in which phase change occurs at
a fixed temperature (Goodrich, 1982). This model does not use a SFCC, and instead, the soil is represented as homogeneous
layers with distinct frozen and thawed thermal properties.

## A7 Hydrus 1D

Hydrus 1D includes a module to simulate water flow and heat transport in frozen soil. The governing equation is written using
the apparent heat capacity formulation and Picard iteration is used to linearize the algebraic nonlinear system. In their paper,
Hansson et al. (2004) explain that during the Picard iteration the solution can easily oscillate whenever the temperature decrease
below the melting temperature. To avoid these oscillation the temperature is reset to the critical value and iteration restarted.
Hydrus 1D adopts an empirical time-step adaptation criterion. It is worthwhile to notice that the modified Picard iteration was
proposed by Celia et al. (1990) to solve the Richards equation – problem for which the NCZ algorithm was originally proposed
(Casulli and Zanolli, 2010).

## A8  MarsFlo

MarsFlo is a three-phase numerical model to simulate the heat transfer and water flow in partially frozen, partially saturated porous media (Painter, 2011). The heat equation is written in the conservative form. The equation is solved using an implicit finite difference method, and the resulting nonlinear system is solved using a Newton-Raphson method. To overcome convergence and stability problems, three modification were introduced (Painter, 2011). There is no mathematical proof that this modified Newton-Raphson algorithm converges.

## A9  NEST

Zhang et al. (2003) developed a one-dimensional physically based model of Northern Ecosystem Soil Temperature (NEST). The heat equation is written in the source term formulation and solved with the DECP approach. The numerical method is explicit in time, thus the maximum time step is of 30 minutes to prevent oscillations in the solution.

## A10  Sergueev et al.

This is a two dimensional model and the governing equation is written in the enthaply form (Sergueev et al., 2003). This model implements a fractional time step approach (Godunov splitting): each time step is divided into two steps and at each step, a different dimension is treated implicitly. The system of finite difference equations is non-linear and is solved with the Newton's method. As in GIPL-2.0, the time derivative of enthalpy is computed either using the difference derivative or the analytical derivative according with the gradient of the temperature field.

## A11  SoilVision

The heat equation is written using the apparent heat capacity. The equation are solved using a finite element solver, FlexPDE suite, both explicit and implicit in time. In case of implicit methods, the resulting non-linear system is solved using the Newton-Raphson method. In the presence of nonmonotic functions, the Newton-Raphson method may fail to converge to the exact solution.

## A12  SUTRA

SUTRA is an established USGS groundwater flow and coupled transport model (Voss and Provost, 2002). McKenzie et al. (2007) and McKenzie and Voss (2013) have extended the model to simulate freezing and thawing processes in the soil. The heat equation is written using the apparent heat capacity formulation and nonlinearities are solved using Picard iteration. Picard iteration does not guarantee to converge to the exact solution.

## A13  Crocus

Crocus is a one-dimensional finite difference model that solves the mass and energy balance within the snowpack taking into account metamorphism and settling. The first versions of Crocus (Brun et al., 1989, 1992) were not enthalpy-based. The

governing equation was written in terms of temperature and water content. It was solved by using the Crank-Nicholson method, and the phase change is treated by using the DECP approach (Brun et al., 1992). After the integration within SURFEX (Vionnet et al., 2012), Crocus uses the enthalpy formulation and the numerical scheme is fully implicit, based on the numerics of ISBA-ES (Boone and Etchevers, 2001). Similarly to the previous version, the heat balance equation is solved adopting the DECP approach (Boone and Etchevers, 2001). Even though recent work Crocus is based on a simple bucket approach for liquid water percolation (Morin et al., 2012; Lafaysse et al., 2017), D'Amboise et al. (2017) implemented a routine for water flow in the snowpack based on the Richards equation, which is characterized by nonlinear behaviour like the enthalpy equation. To solve it, they adopted an approach based on Picard iteration with variable time steps (Paniconi and Putti, 1994).

## A14 SNOWPACK

SNOWPACK (Lehning et al., 1999) solves the heat transfer and creep/settlement equations using a Lagrangian finite element method. The governing equation is written using the source/sink formulation and it is solved using the DECP approach (Bartelt and Lehning, 2002; Lehning et al., 1999). Regarding the water flow, SNOWPACK implements three different schemes: a simple bucket-type approach, an approximation of Richards equation, and the full Richards equation (Wever et al., 2014). The full Richards equation is solved using Picard iteration with variable time steps (Paniconi and Putti, 1994).

## A15 ORCHIDEE

ORCHIDEE is terrestrial biosphere model and it is part of the IPSL-CM4 Earth system model developed by the Institute Pierre Simon Laplace (IPLS) (Krinner et al., 2005). In the version 1.9.6 the snow is described with a single layer of constant density (Wang et al., 2013). Because of the limitations of the this approach, Wang et al. introduce a three-layer snow model, ORCHIDEE-ES, largely inspired from ISBA-ES (Boone and Etchevers, 2001) to consider snow settling, water percolation, and water refreezing. The governing equation is written in the non-conservative form and does not include the latent heat term. The temperature profile is calculated adopting the DECP approach.

## A16 JSBACH

JSBACH is the land surface model developed by the Max Plank Institute (Ekici et al., 2014). It is a component of the Earth System Model (MPI-ESM) that also include ECHAM6 for the atmosphere and MPI-OM for the ocean. JSBACH simulates both the frozen soil and the snowpack. In both cases the heat conduction is assumed to be the dominant method of heat transfer. The governing equation is written in the source term formulation and solved with the DECP approach (Ekici et al., 2014).

## A17 Ice-sheet models

For glacier and ice-sheet models it is necessary to distinguish between cold and temperate ice. Following Aschwanden and Blatter (2005), "ice is treated as temperate if a change in heat content leads to a change in liquid water content alone, and is considered cold if a change in heat content leads to a temperature change alone." This means that cold ice is always below the

melting temperature and thus the phase change does not occur. As result, present-day ice sheet models can be classified into: 'cold-ice method' models and polythermal models.

'Cold-ice method' does not consider the phase change of ice. Because of this the heat capacity can be assumed to be constant and therefore the governing equation can be written in terms of only temperature. These models are easy to implement, but their applicability is restricted since in general temperate zones can be present (Aschwanden and Blatter, 2009). In fact, since the phase change of ice is overlooked, locally, the 'cold-ice method' violates the energy conservation, overestimates the temperate region (Aschwanden and Blatter, 2009), and can not quantify the liquid water content that affects viscosity in temperate ice (Lliboutry and Duval, 1985).

By contrast, polythermal ice-sheet models consider the phase change of ice. Similar to freezing soil models, the polythermal ice-sheet models can be classified in two groups on the base of the treatment of the phase change: front tracking method and enthalpy method (Nedjar, 2002). SICOPLOIS (Greve, 1997a, b; Greve and Blatter, 2016) is the only 'truly' polythermal ice sheet model. It employs the polythermal two-layer scheme (Greve, 1997b): the temperature field and the water content field are computed separately for the ice and temperate domain and a Stefan-type condition is applied at the cold-temperate surface (CTS). This model defines the CTS for both energy flux and mass flux. The drawback of this method relate to the implementation and restriction on the geometry and topology of the CTS (Aschwanden et al., 2012).

Aschwanden and Blatter (2009) presented an enthalpy gradient method. This is a fixed-grid method that differs from the enthalpy method commonly used for freezing soil in its definition of the energy flux. In the enthalpy method, the heat flux is expressed in terms of the temperature gradient, whereas in the enthalpy gradient method it is expressed in terms of enthalpy, assuming that the heat capacity is constant (Aschwanden and Blatter, 2009). The enthalpy approach combines the advantage of solving one equation for the entire domain, cold-ice models, and the correct description of the thermodynamics of temperate ice (front tracking model). This model is implemented in COMSOL Multiphysics (Aschwanden and Blatter, 2009), where nonlinear problems are solved using either a Newton algorithm or a damped Newton algorithm. Also in this case the NCZ may represent a valid option to solve the nonlinear system. To the authors' knowledge, the enthalpy gradient method has not be used in freezing soil models.

Hewitt and Schoof (2016) presented an enthalpy-based finite volume method for polythermal ice. To solve the equation at each time step the computational domain is explicitly divided in the cold and temperate regions, and the energy equation is solved adopting a combination of implicit and explicit methods (Hewitt and Schoof, 2016). It is worth to note that in the temperate region, temperature is set equal to the melting temperature of the ice. This limits the application of this model to simulate freezing soil, where temperature can be larger than the melting temperature of water.

## Appendix B: Pseudocode

We present the pseudocode for a one-dimensional implementation of the NCZ algorithm. Since the matrix $\mathbf{A}$ in Eq. (16) is tri-diagonal we can efficiently compute only the non-zero diagonal: the upper diagonal, the main diagonal, and the lower diagonal. We use the generic expression *Discretize the governing equation* since here, we can choose to adopt either a finite

volume method, as presented in this paper, a finite element method, or a finite difference method. Moreover, the matrix $\mathbf{A}$ is symmetric and positive definite thus within the nested Newton algorithm the linearized algebraic system can be easily solved with the Thomas algorithm. Here, it is worthwhile to point out that when we move to the two-dimensional or three-dimensional problem, the linearized algebraic system cannot be solved with the Thomas algorithm as the matrix is no longer tri-diagonal. In these cases, iterative schemes such as the Conjugate Gradient Method need to be used (Shewchuk, 1994).

---

**Algorithm 1** Program flow

---

1: Read inputs
2: **for** $time = 1, 2, \ldots$ **do**
3:     read boundary conditions
4:     compute enthalpy and thermal conductivity at time level $time$
5:     Discretize the governing equation
6:     **for** $i = 1, 2, \ldots, N_v$ **do**
7:         Compute $\mathbf{A}$ and $\boldsymbol{b}$ of the system applying the boundary condition when $i == 1$ and $i == N_v$
8:     **end for**
9:
10:     Solve the non-linear system: nested Newton algorithm
11:     Set $\boldsymbol{T}^0 < \boldsymbol{T}^*$
12:     **for** $k = 1, 2, \ldots,$ outer iteration **do**
13:         **for** $l = 1, 2, \ldots,$ inner iteration **do**
14:             Solve $(\boldsymbol{P}^{k,l-1} + \mathbf{A} + \boldsymbol{Q}^{k-1})\boldsymbol{T}^{k,l} = \boldsymbol{f}^{k,l-1}$
15:             **if** $\|\boldsymbol{r}^{k,l}\| < \varepsilon$ **then**
16:                 Set $\boldsymbol{T}^k = \boldsymbol{T}^{k,l}$ and exit
17:             **end if**
18:         **end for**
19:         **if** $\|\boldsymbol{r}^k\| < \varepsilon$ **then**
20:             Set $\boldsymbol{T} = \boldsymbol{T}^k$ and exit
21:         **end if**
22:     **end for**
23: **end for**

---

### Appendix C: Fully implicit or semi-implicit formulation, the problem of the thermal conductivity

Using a fully implicit formulation, the discretization of Eq. (1) reads as

$$h_i(T_i^{n+1}) = h_i(T_i^n) + \Delta t \left[ \sum_{j \in \mathcal{F}_i} \Lambda_j^{n+1} \frac{T_{\mathcal{P}(i,j)}^{n+1} - T_i^{n+1}}{\delta_j} + S_i^{n+1} \right] \tag{C1}$$

where $T_i$ is the temperature of the $i$th control volume, $\mathcal{P}(i,j)$ is the neighbour of volume $i$ that shares face $j$ with the $i$th control volume, $\delta_j$ is the nonzero distance between the centers of two adjacent volumes which share the $j$th internal face, $\Delta t$ is the time step size,

$$\Lambda_j^{n+1} = \mathcal{A}_j \max\left[\lambda_i(T_i^{n+1}), \lambda_{\mathcal{P}(i,j)}(T_{\mathcal{P}(i,j)}^{n+1})\right] \tag{C2}$$

and

$$S_i^{n+1} = \int_{\Omega_i} S \, d\Omega \tag{C3}$$

is an optional source/sink term in volume, and $h_i(T)$ is the $i$th enthalpy given by

$$h_i(T) = \int_{\Omega_i} h(T) \, d\Omega \tag{C4}$$

For given initial condition $T_i^0$, at every time step $n = 1, 2, \ldots$, Eq. (C1) constitutes a fully nonlinear system of equation to be solved for $T_i^{n+1}$. To solve Eq. (C1), one sets $T_i^{n+1,0} = T_i^n$. Then the Picard iterations are taken to be

$$h_i(T_i^{n+1,m+1}) = h_i(T_i^n) + \Delta t \left[\sum_{j \in \mathcal{F}_i} \Lambda_j^{n+1,m} \frac{T_{\mathcal{P}(i,j)}^{n+1,m+1} - T_i^{n+1,m+1}}{\delta_j} + S_i^{n+1,m}\right] \tag{C5}$$

where the index $m$ refers to the Picard iteration. By using the NCZ algorithm local and global energy conservation is enforced at each Picard iteration. However, convergence of the Picard iterations is not essential to get a conservative solution for Eq. (1), but some iterations can be used to update the thermal conductivity to the $n+1$th time level Casulli and Zanolli (2010).

## Appendix D: Enthalpy and internal energy

Following the work by Dall'Amico (2010), the internal energy in its canonical form, $U_c$, can be written as

$$U_c = U_c(S, V, M) \tag{D1}$$

where $S$ is the entropy, $V$ is the volume, and $M$ the mass of the constituents. These are the independent variables and are called extensive variables since they depend linearly on the mass of the substance. The first differential of Eq. (D1) is

$$dU_c = \left(\frac{\partial U_c}{\partial S}\right) dS + \left(\frac{\partial U_c}{\partial V}\right) dV + \left(\frac{\partial U_c}{\partial M}\right) dM \tag{D2}$$

According to Callen (1985) it is possible to define

$$\left(\frac{\partial U_c}{\partial S}\right) \equiv T, \text{ the temperature} \tag{D3}$$

$$-\left(\frac{\partial U_c}{\partial V}\right) \equiv p, \text{ the pressure} \tag{D4}$$

$$\left(\frac{\partial U_c}{\partial M}\right) \equiv \mu, \text{ the chemical potential} \tag{D5}$$

With this notation, Eq. (D2) becomes

$$dU_c = TdS - pdV + \mu dM \tag{D6}$$

By making use of the Legendre transformation it is possible to define the enthalpy potential $H_c$ as

$$H_c(S, p, M) = U_c(S, V, M) + pV(S, p, M) \tag{D7}$$

The differential of the enthalpy is

$$dH_c = d[U_c + pV] = TdS - pdV + \mu dM + Vdp + pdV = TdS + \mu dM + Vdp \tag{D8}$$

If we assume that the transformation occurs at constant pressure and volume then Eq. (D6) becomes

$$dU_c = TdS + \mu dM \tag{D9}$$

and Eq. (D8)

$$dH_c = TdS + \mu dM \tag{D10}$$

Hence, from Eq. (D9) and Eq. (D10) the differential of the internal energy and the differential of enthalpy are equal. Therefore the governing equation, Eq. (1), can be equivalently written in term of either the specific enthalpy or the specific internal energy.

**Appendix E: Neumann analytical derivation**

In this section we report the derivation of the Neumann analytical. The enthalpy is defined as

$$h(T) = \begin{cases} \rho_w c_w (T - T_{ref}) + \rho_w l_f & \text{if } T \geq T_m \\ \rho_i c_i (T - T_{ref}) & \text{if } T < T_m - \epsilon \\ \rho_i c_i (T - \epsilon - T_{ref}) + h^{'}(T - (T_m - \epsilon)) & \text{otherwise} \end{cases} \tag{E1}$$

where the singularity of the enthalpy function at $T = T_m$ has been linearized with

$$h^{'} = \frac{\rho_w c_w (T_m - T_{ref}) + \rho_w l_f - \rho_i c_i (T_m - \epsilon - T_{ref})}{\epsilon} \tag{E2}$$

and $\epsilon$ is a parameter defining the temperature range over which the phase change of water occurs, Fig. (E1). In the following tests $\epsilon$ is set to be equal to $0.0001$ °C. The introduction of this linearization is necessary since the enthalpy function needs to be continuously differentiable according to assumption C1 in Section 3.2. It is worth to underline that the temperature range $\epsilon$

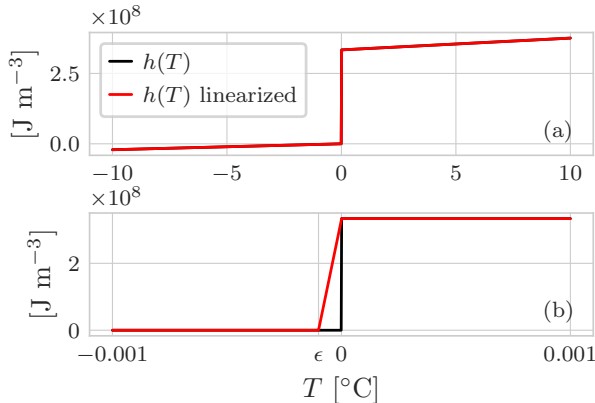

**Figure E1.** (a) Comparison between the enthalpy function of pure water and the enthalpy function used in the numerical model. (b) Note that the energy jump due to the latent heat at $T_m = 0$ °C has been linearized and the steepness is controlled by the parameter $\epsilon$.

can be chosen sufficiently small in order to make this approximation negligible when compared to the physical behaviour of water, considering that: (a) The melting of water in temperate ice is known to actually occur progressively below 0 °C along grain boundaries Langham (1974); Nye and Frank (1973). (b) Freezing often occurs below the melting point when nucleation is relevant. (c) In porous media such as soil, ice melts across a range of temperatures due to the Gibbs-Thompson effect in pores and surface affects at the interfaces between ice and particles (Rempel et al., 2004; Watanabe and Mizoguchi, 2002).

Even though the internal energy function is very steep, the code used does not suffer of convergence problem with a time step of 3600 s. The thermal conductivity is defined as:

$$\lambda(T) = \begin{cases} \lambda_w & \text{if } T \geq T_m \\ \lambda_i & \text{if } T < T_m \end{cases} \tag{E3}$$

Defining the following constant:

$$\alpha_w = \frac{\lambda_w}{\rho_w c_w} \quad \alpha_i = \frac{\lambda_i}{\rho_i c_i} \tag{E4}$$

$$A = \frac{T_m - T_s}{\text{erf}(\gamma)} \quad B = \frac{T_m - T_0}{\text{erf}\left(\gamma\sqrt{\frac{\alpha_i}{\alpha_w}}\right)} \tag{E5}$$

the moving boundary function is

$$\zeta(t) = 2\gamma\sqrt{\alpha_i t} \quad \text{for } t > 0 \tag{E6}$$

where the coefficient $\gamma$ can be found solving the following equation

$$\gamma\sqrt{\alpha_i}l_f\rho - \frac{\lambda_i}{\sqrt{\pi\alpha_i}}Ae^{-\gamma^2} - \frac{\alpha_w}{\sqrt{\pi\alpha_w}}Be^{\gamma^2\frac{\alpha_i}{\alpha_w}} = 0 \tag{E7}$$

**Table E1.** Input parameters for the comparison between Neumann analytical solution and the numerical solution with the NCZ algorithm.

| Symbol | Parameter | Value | Unit |
|--------|-----------|-------|------|
| $\Delta t$ | time step | 60, 300, 3600 | s |
| $\Delta z$ | control volume size | 0.001, 0.005, 0.01 | m |
| $l_f$ | latent heat of fusion | 333700 | J kg$^{-1}$ |
| $c_w$ | specific heat capacity of water | 4187 | J m$^{-3}$ °C$^{-1}$ |
| $c_i$ | specific heat capacity of ice | 2108 | J m$^{-3}$ °C$^{-1}$ |
| $\rho_w$ | water density | 1000 | kg m$^{-3}$ |
| $\rho_i$ | ice density | 970 | kg m$^{-3}$ |
| $\lambda_w$ | thermal conductivity of water | 0.6 | W m$^{-1}$ °C$^{-1}$ |
| $\lambda_i$ | thermal conductivity of ice | 2.09 | W m$^{-1}$ °C$^{-1}$ |
| $\epsilon$ | melting temperature range | 0.0001 | °C |
| $T_0$ | initial temperature | −5, +5 [a] | °C |
| $T_s$ | surface temperature | +5, −5 [a] | °C |

[a] We tested the code both for the freezing case and the thawing case. The thawing case is reported in the Appendix. The first value refers to the freezing case and the second one to the thawing case.

Finally the analytical solution for problem with Dirichlet boundary condition for the thawed and frozen zones are:

$$
\begin{cases}
T(z,t) = T_s + \dfrac{T_m - T_s}{\mathrm{erf}(\gamma)}\,\mathrm{erf}\left(\dfrac{z}{2\sqrt{\alpha_i t}}\right) & 0 < z < \zeta(t) \\[4mm]
T(z,t) = T_0 + \dfrac{T_m - T_0}{\mathrm{erfc}\left(\gamma\sqrt{\dfrac{\alpha_i}{\alpha_w}}\right)}\,\mathrm{erfc}\left(\dfrac{z}{2\sqrt{\alpha_w t}}\right) & z > \zeta(t)
\end{cases}
\tag{E8}
$$

## Appendix F: Lunardini analytical derivation

The solution fo the Lunardini problem (i.e. the Lunardini solution) as described by McKenzie (McKenzie et al., 2007) is given by the following set of equations:

$$
T_1 = (T_m - T_s)\frac{\mathrm{erf}\left(\dfrac{x}{2\sqrt{\alpha_1 t}}\right)}{\mathrm{erf}(\psi)} + T_s
\tag{F1}
$$

$$
T_2 = (T_m - T_f)\frac{\mathrm{erf}\left(\dfrac{x}{2\sqrt{\alpha_4 t}}\right) - \mathrm{erf}(\gamma)}{\mathrm{erf}(\gamma) - \mathrm{erf}\left(\psi\sqrt{\dfrac{\alpha_1}{\alpha_4}}\right)} + T_f
\tag{F2}
$$

$$T_3 = (T_0 - T_f)\frac{-\operatorname{erfc}\left(\dfrac{x}{2\sqrt{\alpha_3 t}}\right)}{\operatorname{erfc}\left(\psi\sqrt{\dfrac{\alpha_4}{\alpha_3}}\right)} + T_0 \tag{F3}$$

where $T_1$, $T_2$, and $T_3$ are the temperatures at distance, $x$, from the temperature boundary for the frozen, partially frozen, and unfrozen zone respectively; $erf$ and $erfc$ are the error function, and the complementary error function respectively; $T_0$, $T_m$, $T_f$, and $T_s$ are the temperatures of the initial condition; the solidus, the liquidus, and the boundary temperature, respectively; $\alpha_1$ and $\alpha_3$ are the thermal diffusivities for the frozen, and unfrozen zone respectively, defined as $\lambda_1/C_1$ and $\lambda_3/C_3$ where $C_1$ and $C_3$ are the volumetric bulk-heat capacities of the frozen and unfrozen zones. The thermal diffusivity of the partially frozen zone is assumed to be constant across the transition region, and the thermal diffusivity with latent heat term included, $\alpha_4$, is defined as:

$$\alpha_4 = \frac{\lambda_2}{C_2 + \dfrac{\gamma_d l_f \Delta\xi}{(T_f - T_m)}} \tag{F4}$$

where $\gamma_d$ is the dry unit of soil solids, and $\Delta\xi = \xi_0 - \xi_f$ where $\xi_0$ and $\xi_f$ are the ratio of unfrozen water to soil solids for the fully thawed and frozen conditions respectively. For a time, $t$, in the region $0 \leq x \leq X_1(t)$ the temperature is $T_1$ and $X_1(t)$ is given by

$$X_1(t) = 2\psi\sqrt{\alpha_1 t} \tag{F5}$$

and from $X_1(t) \leq x \leq X(t)$ the temperature is $T_2$, where $X(t)$ is given by

$$X(t) = 2\gamma\sqrt{\alpha_4 t} \tag{F6}$$

and for $x \geq X(t)$ the temperature is $T_3$. The unknowns, $\psi$ and $\gamma$, are solving the set of these two equations:

$$\frac{T_m - T_s}{T_m - T_f}\exp^{-\psi^2(1-\alpha_1/\alpha_4)} = \frac{\dfrac{\lambda_2}{\lambda_1}\sqrt{\dfrac{\alpha_1}{\alpha_4}}\operatorname{erf}(\psi)}{\operatorname{erf}(\gamma) - \operatorname{erf}\left(\psi\sqrt{\dfrac{\alpha_1}{\alpha_4}}\right)} \tag{F7}$$

$$\frac{(T_m - T_f)\dfrac{\lambda_2}{\lambda_3}\dfrac{\alpha_3}{\alpha_4}}{T_m - T_f}\exp^{-\gamma^2(1-\alpha_4/\alpha_3)} = \frac{\operatorname{erf}(\gamma) - \operatorname{erf}\left(\sqrt{\dfrac{\alpha_1}{\alpha_4}}\psi\right)}{\operatorname{erf}\left(\gamma\sqrt{\dfrac{\alpha_4}{\alpha_3}}\right)} \tag{F8}$$

## Appendix G: Numerical test

s explained in Section 5, comparing the position of the zero-isotherm after 100 years using three different time step, hourly, daily and 10-days, there are no significant deviation in the results. The larger deviation occur when the zero-isotherm is

**Table F1.** Input parameters for the comparison between Lunardini analytical solution and the numerical solution with the NCZ algorithm.

| Symbol | Parameter | Value | Unit |
|---|---|---|---|
| $\Delta t$ | time step | 300, 900, 3600 | s |
| $\Delta x$ | control volume size | 0.01 | m |
| $L_f$ | latent heat of fusion | 334560 | $\text{J kg}^{-1}$ |
| $C_1$ | volumetric heat capacity, frozen | 690030 | $\text{J m}^{-3}\,{}^\circ\text{C}^{-1}$ |
| $C_2$ | volumetric heat capacity, partially frozen | 690030 | $\text{J m}^{-3}\,{}^\circ\text{C}^{-1}$ |
| $C_3$ | volumetric heat capacity, unfrozen | 690030 | $\text{J m}^{-3}\,{}^\circ\text{C}^{-1}$ |
| $\gamma_d$ | dry unit density of soil solids | 1680 | $\text{kg m}^{-3}$ |
| $\xi_0$ | ratio of liq. water to soil solids, unfrozen | 0.2 | - |
| $\xi_f$ | ratio of liq. water to soil solids, frozen | 0.0782 | - |
| $\lambda_1$ | thermal conductivity, frozen | 3.462696 | $\text{W m}^{-1}\,{}^\circ\text{C}^{-1}$ |
| $\lambda_2$ | thermal conductivity, partially frozen | 2.939946 | $\text{W m}^{-1}\,{}^\circ\text{C}^{-1}$ |
| $\lambda_3$ | thermal conductivity, unfrozen | 2.417196 | $\text{W m}^{-1}\,{}^\circ\text{C}^{-1}$ |
| $\gamma$ | solution parameter for Eq. (F7) and Eq. (F8) | 5.616, 2.060, 1.397 [a] | - |
| $\psi$ | solution parameter for Eq. (F7) and Eq. (F8) | 0.158, 0.137, 0.061 [a] | - |
| $T_0$ | initial temperature | +4 | °C |
| $T_s$ | boundary temperature | −6 | °C |
| $T_f$ | liquidus temperature | 0 | °C |
| $T_f$ | solidus temperature | −0.1, −1, −4 | °C |

[a] The first value refers to $T_m = -0.1\,{}^\circ\text{C}$ the second value t$T_m = -1\,{}^\circ\text{C}$, and the third value to $T_m = -4\,{}^\circ\text{C}$.

shallow: at the beginning of the thawing season as well as the freezing one. At the begging of the thawing season, Fig. (G1), there is a time lag of about one month between the beginning of the thawing season for the hourly simulation and the 10-days simulation. This can be attributed to different surface temperature used to drive the simulations. In particular, in the case of the hourly simulation it is possible to see the oscillations of the position of the zero-isotherm, panel (c), related to the oscillation of the surface temperature around 0 °C, panel (a). Figure (G2) shows the detail of the freezing season. In panel (c) it is possible to note that when the zero-isotherm is deep there is a good agreement between the three simulations. The main differences occurs at the soil surface since with larger time steps the signal of the surface boundary condition is smoothed and does not oscillates around 0 °C. Moreover, by using an hourly time step and a daily time step it is possible to capture the joining of the downward and upward freezing front, while this is not possible with the 10-days time step since the joining occurs in-between of two consecutive time step.

As explained in Section 5, the maximum temperature profile, Fig. (12) panel (d), presents an 'elbow' due to the so-called zero curtain effect. The zero curtain effect, Fig. (G3), is the period of time during which the temperature remains nearly constant and very close to the freezing point because of the latent heat released during the phase change of water.

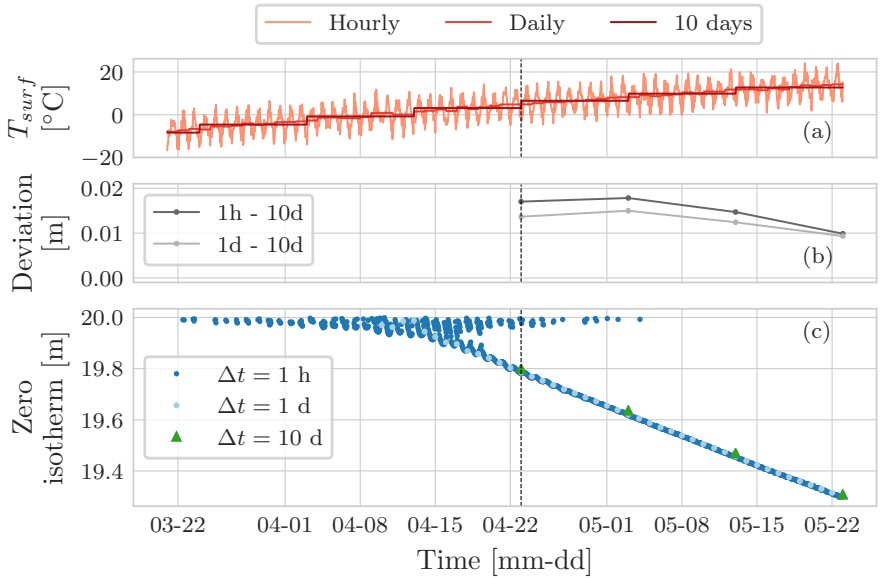

**Figure G1.** Detail of the beginning of the thawing season for the year 1999. Panel (a) shows the surface temperature for the hourly, the daily and the 10-days simulations. Panel (b) shows the deviation of the position of the zero-isotherm after 100 years between the hourly and the 10-days simulation, and between the daily and the 10-days simulation. Panel (c) shows the position of the zero-isotherm after 100 years for the three simulations. In (b) there is a time lag of about one month between the beginning of thawing season for the hourly simulation and the 10-days one, dashed grey line. This can be attributed to the different surface temperature used to drive the simulations.

Figure (G4) shows the temperature envelopes for the hourly, the daily, and the 10-days simulations setting the thermal conductivity of water equal to the thermal conductivity of ice, $\lambda_w = \lambda_i$. It is interesting to note that mean temperature, panel (c), is constant throughout the soil column. The mean temperature is very close to the initial temperature profile that is also equal to the mean surface boundary condition.

Figure (G5) shows a comparison of the the zero-isotherm position by using a coarser space discretization. Again, the three simulation with the hourly time step, the daily time step, and the 10-days time step are still in good agreement. By using a coarser spatial discretization, the zero-isotherm, panel (c), presents some 'steps' that are not present when using a finer grid, Fig. (11). Moreover, the joining of the downward and upward freezing front is not captured neither by the hourly nor by the daily simulations.

For this numerical test we checked the mean number of iterations required to solve the nonlinear system with the NCZ algorithm, the Newton-Raphson algorithm, and the globally convergent Newton algorithm. We performed a simulation lasting 1 year with a time step $\Delta t = 1$ h and for different spatial discretizations. As can be seen in Table G2, neither the Newton-Raphson nor the globally convergent Newton converge: they always reach the maximum number of iterations allowed with a consequent increase of the computational cost.

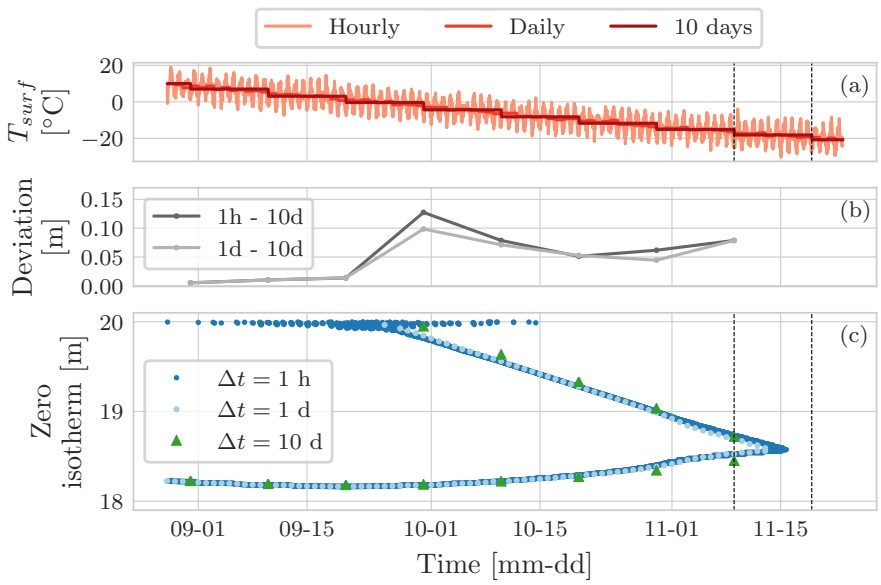

**Figure G2.** Detail of the beginning of the freezing season for the year 1999. Panel (a) shows the surface temperature for the hourly, the daily and the 10-days simulations. Panel (b) shows the deviation of the position of the zero-isotherm after 100 years between the hourly and the 10-days simulation, and between the daily and the 10-days simulation. Panel (c) shows the position of the zero-isotherm after 100 years for the three simulations. The joining of the downward and upward freezing front is captured by the hourly and the daily simulations, (c). It is interesting to note that for the 10 days simulation the joining occurs in-between of two consecutive time step.

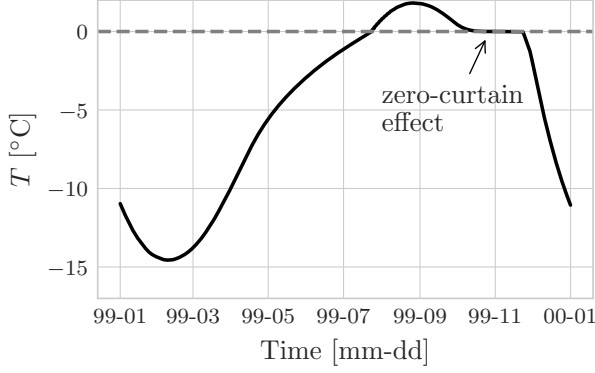

**Figure G3.** Hourly temperature at 1.5 m depth for the year 1999. Note the prolonged period of 43 days (11 October until 23 November) when temperature remained within ±0.1 °C. This is the so-called zero-curtain effect, and it is due to latent heat of fusion that is continually released during the freezing of soil moisture.

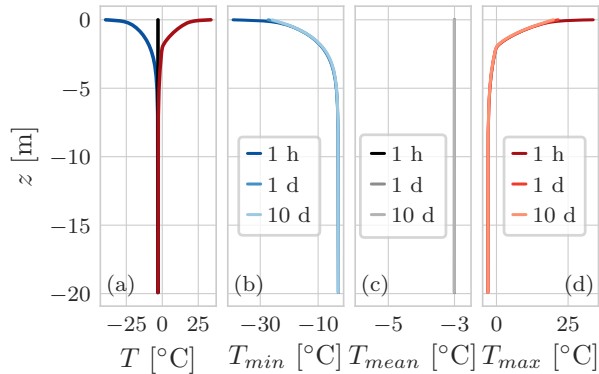

**Figure G4.** Temperature profile envelope considering $\lambda_w = \lambda_i$. (a) The minimum, mean, and maximum temperature profile for the hourly simulation. (b), (c), (d) show the comparison of the minimum, mean, and maximum temperature profile respectively for the three simulations: with an hourly surface temperature boundary condition and $\Delta t = 1$ h, with a daily surface temperature boundary condition and $\Delta t = 1$ day, with a ten day surface temperature boundary condition and $\Delta t = 10$ day. All three simulations last 100 years. Because $\lambda_w = \lambda_i$ the mean temperature, panel (c), is constant throughout the soil column and it is not possible to appreciate the thermal offset. The mean temperature is very close to the initial temperature profile, the maximum error is of $0.003$ °C.

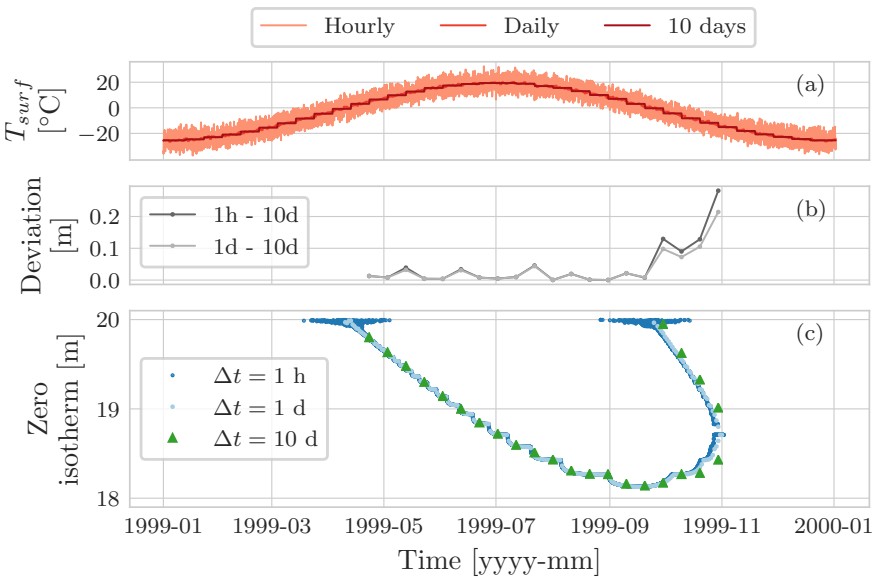

**Figure G5.** Comparison of the position of the zero-isotherm, panel (c), after 100 years of three simulations: using an hourly boundary condition with time step of $\Delta t = 1$ h, using a daily boundary condition with a time step of $\Delta t = 1$ day, and a 10-day boundary condition with a time step of $\Delta t = 10$ day. Panel (a) shows the surface temperature for the hourly, the daily and the 10-days simulations. Panel (b) shows the deviation of the position of the zero-isotherm after 100 years between the hourly and the 10-days simulation, and between the daily and the 10-days simulation. By using a coarser spatial discretization, the zero-isotherm presents some 'steps', panel (c), independently on the size of the time step. Another consequence of this is that the joining of the downward and upward freezing front is not captured neither by the hourly nor by the daily simulations.

**Table G1.** Input parameters for the numerical tests.

| Symbol | Parameter | Value | Units |
|---|---|---|---|
| $\Delta t$ | time step | 3600, 86400, 864000 | s |
| $\Delta z_{min}^a$ | thickness of the first control volume | 0.002, 0.005 | m |
| $b^a$ | growth rate ground depth | 0.01, 0.1 | – |
| $z_{max}$ | maximal ground depth | 20 | m |
| $l_f$ | latent heat of fusion | 333700 | $J\,kg^{-1}$ |
| $c_w$ | specific heat of water | 4188 | $J\,m^{-3}\,°C^{-1}$ |
| $c_i$ | specific heat of ice | 2117 | $J\,m^{-3}\,°C^{-1}$ |
| $c_{sp}$ | specific heat of soil particles | 1000 | $J\,m^{-3}\,°C^{-1}$ |
| $\rho_w$ | water density | 1000 | $kg\,m^{-3}$ |
| $\rho_i$ | ice density | 1000 | $kg\,m^{-3}$ |
| $\rho_{sp}$ | soil particles density | 2700 | $kg\,m^{-3}$ |
| $\lambda_w$ | thermal conductivity of water | 0.6 | $W\,m^{-1}\,°C^{-1}$ |
| $\lambda_i$ | thermal conductivity of ice | 2.09 | $W\,m^{-1}\,°C^{-1}$ |
| $\lambda_{sp}$ | thermal conductivity of soil particles | 3.0 | $W\,m^{-1}\,°C^{-1}$ |
| $\theta_s$ | saturation water content | 0.46 | – |
| $\theta_r$ | residual water content | 0.1 | – |
| $\alpha$ | Van Genuchten parameter | 1.5 | $m^{-1}$ |
| $n$ | Van Genuchten parameter | 1.2 | – |
| $T_0$ | initial temperature | $-3$ | °C |
| | SFCC | Dall'Amico | |
| | Thermal conductivity model | Johansen | |

[a] We used two different space discretizations. The thickness of the ground layer is parametrized as
$dz_i = dz_{min}(1+b)^{(i-1)}$ (Gubler et al., 2013).

**Table G2.** Summary of the mean number of iterations for the NCZ algorithm, the Newton-Raphson algorithm (N. R.), and the globally convergent Newton algorithm (g. c. N.). The simulation lasts $1$ h with a time step $\Delta t = 1$ day. We considered different spatial discretizations. The tolerance $\varepsilon = 10e - 11$ has been rescaled with the water latent heat of fusion and the water density. The maximum number of iteration for each time step is $40$. As can be seen the Newton-Rapshon and the globally convergent Newton does not converge so it always reaches the maximum number of iteration allowed.

|  | # control volumes 500 | # control volumes 1000 | # control volumes 2000 | # control volumes 5000 | # control volumes 10000 |
|---|---|---|---|---|---|
| Mean number of iterations NCZ | 12 | 13 | 14 | 16 | 18 |
| Mean number of iterations N. R. | 40 | 40 | 40 | 40 | 40 |
| Mean number of iterations g. c. N. | 40 | 40 | 40 | 40 | 40 |

*Author contributions.* Conceptualization N.T., S.G., R.R; software N.T., R.R.; writing of original draft N.T.; reviewing and editing N.T., S.G., R.R. All the authors have read and agreed to the published version of the manuscript.

*Competing interests.* The authors declare no conflict of interest.

*Acknowledgements.* The first author would like to thank Professor Vincenzo Casulli and Professor Michael Dumbser of the Department of Civil, Environmental and Mechanics Engineering at the University of Trento for their fruitful discussions on the numerical aspects of the work. We thank Professor Andy Aschwanden, Professor Ed Bueler, and Professor Constantine Khrulev for their feedback regarding ice sheet models, and Dr. Samuel Morin for feedback regarding the model Crocus. We especially thank Dr. Matthieu Lafaysse and the anonymous reviewer for their constructive comments that helped to reasonably improve our manuscript.

This work has been partially supported by a Ph.D. grant by the Department of Civil, Environmental and Mechanics Engineering at the University of Trento. In Canada, support was available through the NSERC Strategic Project "Improved Characterization of Permafrost Vulnerability to Support Decision Makers, Infrastructure, and Community Stewardship in the Northwest Territories" and NSERC PermafrostNet.

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
