# Peer review of "A method for solving heat transfer with phase change in ice or soil that allows for large time steps while guaranteeing energy conservation"

_The Cryosphere, 2020_

## Referee Comment (RC1) · Matthieu Lafaysse (Referee) · 25 Nov 2020

In this paper, the authors present the first application of an iterative algorithm published in 2010 in a mathematics journal to solve the coupled processes of heat diffusion and phase changes (the Stefan problem) in a context easily extendable to any frozen soil, ice or snowpack model. I think this paper may become a major reference for the future generation of models of the cryosphere components. Indeed, I think that the failure of classical algorithms to converge towards stable solutions is poorly known in the community, at least in the snow science community I belong. The innovative character of this paper is indisputable as the references in the literature providing ways to solve

[Figure]

Stefan problems are extremely scarce : the main reference for ice is still Voller (1990 !!), and for snow the problem just seems to be ignored by the community while we recently got proofs that the decoupled treament of current snow models is responsible for significant errors in some situations. The paper has an excellent structure, language and overall quality and it could almost be published as it is. Nevertheless, I have a few comments and suggestions mainly in the idea of better encouraging model developers to consider this approach for their applications by providing all informations they could expect. Nothing is absolutely mandatory but I would be even more enthusiastic if the authors could consider some of these suggestions. In any case, I thank and congratulate the authors for this very high quality paper.

**1 Main comments**

1. My main concern is the fact that it is rather difficult to understand the algorithm only with the material provided in this paper. The introduction of Section 3.2 and the sentences Lines 220-221 remained really obscur to me even after checking Appendix B. Combining the reading with an attentive analysis of Casulli and Zanolli, 2010 finally allowed me to perfectly understand the process, but it asked me significant efforts because (1) the notations differ between both papers, (2) I could not understand why Fig. 1 and Fig. D1 looked inconsistent with the formulations of enthalpy in Eq. 8 and Eq. D1, and (3) there is room for improvement in the presentation of the algorithm in Appendix B.

   - Therefore, I would first strongly suggest to reproduce Eq. 13-19 of Casulli and Zanolli after Eq. 20 of this paper by using the own notations of this paper. Even if the details of the linearization of the two functions in this iterative procedure is already published, I think it would really help the cryosphere modellers to provide again the detailed equations for their

understanding of the numerical approach. I actually had to write all the equations myself to finally understand.

- Then, I think that Fig. 1 and D1 should be modified (either the curves, either the axis legends) because H can not be constant with temperature. Ideally, the plots could also be a bit larger with some trick to better distinguish overlapping curves.

- Then, I am sorry to say that Algorithm 1 in Appendix B is really confusing and that the version detailed by Casulli and Zanolli was much clearer for me. First, a key point is missing i.e. the initialization of the guess of the inner iteration ($T_k^{i,0}$). The comments "linearize $h1$" and "linearize $h2$" are ambiguous because they would suggest that some code instruction is required here whereas it is not the case. $rhs$ is not defined. Is it the $b$ vector of Eq. 20 ? The use of the layer index $i$ in superscript next to the inner iteration index is confusing and putting both iteration indexes at the same level as in Casulli and Zanolli seems a better choice. Furthermore, the layer index $i$ is in subscript in the main text. The meaning of superscript n (previous time step ?) might also be remind. I am not sure if $d_k$ is equivalent to $f^{n,m-1}$ combined with $d^{n-1}$ in Casulli and Zanolli ? If yes (I guess so), I think that separating the terms which do not depend on iteration $m$ would help understand the role of both iterations. Furthermore it is cheaper to compute them outside the inner loop. Therefore, why not keeping the presentation of Casulli and Zanolli on that point with the computation of $d$ in the outer loop and $f$ in in the inner loop ? Finally, it's a detail but the condition on the residuals to exit the loops is not explicitly formalized : is the threshold applied on each element of the vector or on a norm of the vector ?

2. The comparison of numerical results with analytical results is very convincing (Section 4). However, I believe that the comparisons with other numerical approaches is essential if the authors want to convince the cryosphere modelers to change their algorithms. Therefore, I think that the comparison with other numerical models might have found its place in the main text rather than in Appendix G. In the same idea, the comparison presented by the authors with the Newton-Raphson algorithm is not fully representative of the shortcomings of the existing models as the literature review presented by the authors show that a number of models use even worse representations (DECP is the standard for snow models). Of course, I understand that DECP can not be seen as a state-of-the-art reference for the authors, but would it be possible to include it in the comparisons so that modelers using this approach feel more concerned ?

3. Finally, my last main remark is that modelling in geoscience is often a compromise between physical accuracy and numerical cost. This paper pays attention to the accuracy and stability of the solution. The numerical cost is considered through the possibility to extend the time step (section 5). However, in a number of surface models, the time step is often constrained by the need to represent the diurnal cycle and the time resolution necessary for the other processes. Therefore, I would be interested by a discussion about the numerical cost of the different approaches for a fixed time step (typically 10 to 60 minutes for soil or snow models resolving the diurnal cycle). Indeed, the algorithm proposed by the authors requires to solve $k \times m$ linear systems. It means that the improved accuracy (guaranteed convergence and stability) comes with a potentially much more expensive cost than the approaches published by Voller for ice (iterative but without any linear system to solve), or than the DECP approach commonly used in snow models and requiring the solving of only one linear system. In particular, would it be possible to estimate the number of iterations required in the examples provided in the two test cases ? How fast is the convergence in the simple case

of heat diffusion without any phase change ? Is it possible to estimate an average number of iterations in long simulations based on real forcing conditions ?

**2 Minor comments**

Lines 49-55 : It might also be interesting to mention that the notion of interface is also often meaningless in ice and snow where thick isotherm layers of a mixed solid-liquid medium are very common.

Line 74 : Applications with long time steps also include the surface components of climate models and Numerical Weather Prediction models, and also models dedicated to avalanche hazard forecasting.

Line 115 : "SFCC have an inflection point". With just a quick look at Bao et al., 2016, I could not find to which figure or comment this statement refers. Could you provide the details ?

Section 2 is really nice and interesting to read, but a number of symbols definitions come a bit too late after their first use in an equation. For instance, enthalpy already appears in Eq. 1 but is only defined from Eq. 5 to Eq.8. Similarly, latent heat of fusion and liquid water fraction are used in Eq. 3 but are only defined after Eq . 7. It is not critical for the understanding but maybe some reordering could manage to avoid these late definitions.

Eq. 9 : Although their meaning is relatively obvious, please do not forget to define indexes i and n.

[Figure]

Line 179 : I think Eq. 12 corresponds to the implicit case, not semi-implicit, am I wrong ?

Eq. 12 : Source terms seem to be expressed at the beginning of the time step in this formalism. In surface models, I think that it is relatively common to express the source terms which depend on temperature at the end of the time step in order to improve the stability especially because of the longwave surface radiation function of surface temperature. I think this does not affect the possibility to apply this algorithm because when these terms are linearized, it just adds terms in the coefficients of the A tridiagonal matrix and in the b vector without changing the formalism. But maybe the authors could just mention than source terms do not necessarily have to be expressed at the beginning of the time step to allow this algorithm to be applied.

Line 432 It was solved

Lines 436-438 This is true but this development has not really been finalized until now and all recent works using Crocus are still based on a simple bucket approach for liquid water percolation.

Table 1 and Appendix A14 : I don't really see what is the difference between Crocus and SNOWPACK. I do not understand what means "phase change are accounted for as volumetric heat sinks (melting) and sources (refreezing)" and why a nonlinear solver would be "not required". Is SNOWPACK algorithm really different from the DECP approach ? Is it possible to better explain the difference if any ?

General comment about Appendix A13 and A14 : I also think that most multilayer snow schemes simpler than Crocus and SNOWPACK and typically used in climate or hydrology models are also based on this DECP approach. Maybe it could be mentioned somewhere.

**3  Typos**

Line 22 these models

Line 175 orthogonal

Line 312 to match the aim

---

## Referee Comment (RC2) · Anonymous Referee #2 · 30 Nov 2020

**Review: "A method for solving heat transfer with phase change in ice or soil that allows for large time steps while guaranteeing energy conservation"**

by Tubini et al.

Submitted to *The Cryosphere*

**1  General**

In this paper, the authors describe a new numerical model for solving phase change problems with applications to the cryosphere. As the authors correctly identify, phase change is central to snow, ice, permafrost, and other components of frozen landscapes. What I like about this paper is that it uses modern numerical methods. The application of Newton-Casulli-Zanolli (NCZ) method to solve the parabolic pdes that arise in phase change is a great idea. What I don't like about this paper is that (i) it is not clear what cryospheric problem they are attempting to solve, (ii) they ignore relevant literature, and (iii) their pseudocode and code are not in a form that is useful for the broader community. Additionally, I wonder if the *The Cryosphere* is the correct venue – I think that *Geoscience Model Development* would be a much better fit for this paper. As it stands, I am hesitant to support publication in its current form.

**2  Remarks**

1. In the abstract, the authors state that "the nonlinear behaviour of enthalpy as function of temperature can prevent thermal models of snow, ice and frozen soil from converging to the correct solution" but do not provide a description or citation for this claim. Reading further into the paper, this claim is based on a survey of experts, cited as personal communications. I appreciate the point that the authors are trying to make and it is indeed an important advance of the NCZ method, but the phrasing could be improved throughout for clarity.

2. The main advance that this paper reports is the implementation of a novel algorithm for solving phase change problems. However, the pseudocode included in the paper is not very useful as opposed to a well-documented version of the code that is easy to run. While the code is released publicly, the github documentation is unclear, written in java (for good reason), and not approachable. I was hopeful that I could run the code but that did not seem feasible.

3. This paper neglects to cite or engage with Schoof and Hewitt (2016), who derive a general enthalpy model for phase change. Schoof and Hewitt (2016) follows on from Aschwanden and Blatter (2009) and Aschwanden et al. (2012), where only the first paper is referenced in the manuscript. Beyond the numerical implementation of the NCZ method, it is not clear what this paper adds beyond Schoof and Hewitt (2016) in terms of physical understanding and the role of enthalpy in phase change.

**3 Specific comments**

1. I suggest replacing the title with: "An undated numerical method for solving the heat equation with phase change".

2. line 105: is $\theta_s$ defined anywhere?

3. line 115: the kink in the sfcc only matters if the authors take the derivative, which is not required! I am not sure about the value of this 'straw-man' argument about the three 'identical yet different' representations of the heat equation. First off, the language is unclear, so it is opaque as to what method the authors will actually use.

4. line 163: semi-implicit is not required, implicit is required. semi-implicit is a convenient method of mixing explicit and implicit methods to decrease time step restrictions.

5. line 169: ok, let me get this straight: the authors asked their colleagues if there is guaranteed convergence for nonlinear problems using the 'currently used algorithms' and they said no? Tell me more. Tell me why convergence is not guaranteed and how NCZ guarantees it – don't refer me to their paper. That is not the point. All the authors need to say is that NCZ offers advantages. Instead the authors generate an entire table showing that all of the methods they can think of have drawbacks, based on the word of their colleagues? The articulation of this argument needs substantial bolstering.

6. line 195: it looks like it comes down to the fact that the enthalpy is, for some reason, not monotonic with temperature, but isn't that the reason to use the enthalpy: because it is monotonic? I agree that at the melting temperature there is a jump in enthalpy governed by the latent heat, but does that mean that it is not monotonic?

7. line 202: I must be very confused, why don't you just solve for the enthalpy and use the jump conditions to determine the temperature (Schoof and Hewitt, 2016)?

8. section 4: I have no idea what the Neumann and Lunardini solutions are: describe the problems physically? I can certainly look in the appendix (and did) to find the mathematics, but until I saw Figure 2, I was totally confused at what problem you were trying to solve.

9. line 288: SUTRA uses an $\epsilon$ in the enthalpy function as well?

10. table 2: there does not seem to be monotonic convergence. given that this paper is claiming guaranteed convergence, I would have liked to see a convergence plot showing that the solution does converge at the power of the discretization, both in space and time. Also, it is worth mentioning the error order for both, especially since the method is first-order in time! predictor-corrector methods (or Heun's method) could be used instead of Crank-Nicholson to increase the resolution without the same time step restrictions.

11. figure 4: if the point is to show that the left and right panels are the same, then I suggest, plotting them on one panel and using the other panel to show the difference.

12. section 4.2: what defines the mushy zone in the Lunardini analytical solution? and how is this different than Katz (2008)?

13. line 329: is this a paragraph fragment?

14. Most figures: the axis labels as well as figure text are missing letters and difficult to read.

15. line 510: why is $\epsilon$ required? It seems that the value of the enthalpy is that there is a smooth transition across the phase change – adding $\epsilon$ negates the authors' claim that they are 'guaranteeing energy conservation', because they have added a fictious mushy zone.

**References**

A. Aschwanden and H. Blatter. Mathematical modeling and numerical simulation of polythermal glaciers. *J. Geophys. Res.*, 114(F1), 2009. doi: 10.1029/2008JF001028.

A. Aschwanden, E. Bueler, C. Khroulev, and H. Blatter. An enthalpy formulation for glaciers and ice sheets. *J. Glaciol.*, 58(209):441–457, 2012. doi: 10.3189/2012JoG11J088.

R. F. Katz. Magma dynamics with the enthalpy method: Benchmark solutions and magmatic focusing at mid-ocean ridges. *J. Petrol.*, 49(12):2099–2121, 12 2008. ISSN 0022-3530. doi: 10.1093/petrology/egn058.

C. Schoof and I. J. Hewitt. A model for polythermal ice incorporating gravity-driven moisture transport. *J. Fluid Mech.*, 797:504–535, Jun 2016. doi: 10.1017/jfm.2016.251.

---

## Referee Comment (RC3) · Matthieu Lafaysse (Referee) · 3 Dec 2020

I just wanted to mention that in my review, I was actually refering to Casulli and Zanolli, 2012 (Journal of Computational and Applied Mathematics) and not to Casulli and Zanolli, 2010 (SIAM Journal on Scientific Computing) as written in my review. As I use notations and references to the equations of the 2012 paper, it would be confusing to read my comment with the incorrect reference. I am sorry for this mistake, I did not notice that these authors published two disctinct papers on this algorithm.

---

## Author Comment (AC1) · 28 Jan 2021

Dear Dr. Matthieu Lafaysse,

Thank you very much for your review and constructive comments. The entire text of your referee comment is shown (ML) together with our authors' responses (AR).

Kind regards, Niccolò Tubini – on behalf of all authors

**Main comments**

1. **ML**: My main concern is the fact that it is rather difficult to understand the algorithm only with the material provided in this paper. The introduction of Section 3.2 and the sentences Lines 220-221 remained really obscure to me even after checking Appendix B. Combining the reading with an attentive analysis of Casulli and Zanolli, 2010 finally allowed me to perfectly understand the process, but it asked me significant efforts because (1) the notations differ between both papers, (2) I could not understand why Fig. 1 and Fig. D1 looked inconsistent with the formulations of enthalpy in Eq. 8 and Eq. D1, and (3) there is room for improvement in the presentation of the algorithm in Appendix B.

   - Therefore, I would first strongly suggest to reproduce Eq. 13-19 of Casulli and Zanolli after Eq. 20 of this paper by using the own notations of this paper. Even if the details of the linearization of the two functions in this iterative procedure is already published, I think it would really help the cryosphere modellers to provide again the detailed equations for their understanding of the numerical approach. I actually had to write all the equations myself to finally understand.

   **AR**: In the resubmitted manuscript we have reproduced Eq. 13-19 of Casulli and Zanolli, 2010 as you suggested. The notation is slightly different since the quantities in the Richards' equation are different from those of the heat equation. But we tried try to keep the same notation when it was possible.

   - **ML**: Then, I think that Fig. 1 and D1 should be modified (either the curves, either the axis legends) because H cannot be constant with temperature. Ideally, the plots could also be a bit larger with some trick to better distinguish overlapping curves.

   **AR**: About Fig. D1 the problem is that the temperature range is too small to appreciate the variation of h(T) with T because of the magnitude of latent heat. To help the understanding we modified the Fig. D1 as follows: in (a) we have the enthalpy function, and (b) is a detail on the linearization for the latent heat.

[Figure]

As regards Fig. 1, we have reduced the temperature range and increased the size, it is now a two-columns figure. However, the variation of h with T for T > 0 [°C] is not evident because of the magnitude of latent heat and the small temperature range for T > 0 [°C]

[Figure]

- **ML**: Then, I am sorry to say that Algorithm 1 in Appendix B is really confusing and that the version detailed by Casulli and Zanolli was much clearer for me. First, a key point is missing i.e. the initialization of the guess of the inner iteration (Ti,0k). The comments "linearize h1" and "linearize h2" are ambiguous because they would suggest that some code instruction is required here whereas it is not the case. rhs is not defined. Is it the b vector of Eq. 20 ? The use of the layer index i in superscript next to the inner iteration index is confusing and putting both iteration indexes at the same level as in Casulli and Zanolli seems a better choice. Furthermore, the layer index i is in subscript in the main text. The meaning of superscript n (previous time step ?) might also be remind. I am not sure if dk is equivalent to fn,m−1 combined with dn−1 in Casulli and Zanolli ? If yes (I guess so), I think that separating the terms which do not depend on iteration m would help understand the

role of both iterations. Furthermore, it is cheaper to compute them outside the inner loop. Therefore, why not keeping the presentation of Casulli and Zanolli on that point with the computation of d in the outer loop and f in in the inner loop ? Finally, it's a detail but the condition on the residuals to exit the loops is not explicitly formalized: is the threshold applied on each element of the vector or on a norm of the vector?

**AR**: About the Algorithm 1 in Appendix B, I will report the version by Casulli and Zanolli specifying the index and apex. The threshold is applied on the norm of the vector of the residuals.

2. **ML**: The comparison of numerical results with analytical results is very convincing (Section 4). However, I believe that the comparisons with other numerical approaches is essential if the authors want to convince the cryosphere modelers to change their algorithms. Therefore, I think that the comparison with other numerical models might have found its place in the main text rather than in Appendix G. In the same idea, the comparison presented by the authors with the Newton-Raphson algorithm is not fully representative of the shortcomings of the existing models as the literature review presented by the authors show that a number of models use even worse representations (DECP is the standard for snow models). Of course, I understand that DECP can not be seen as a state-of-the-art reference for the authors, but would it be possible to include it in the comparisons so that modelers using this approach feel more concerned?

**AR**: Appendix G has been moved in Section 4.1.
We understand that a comparison with other model would be of interest. However, running other models over long periods as the one we used in this paper would tremendously delay our paper resubmission. For our assessment we rely on literature and communications with the Authors of some of the paper we cite. Besides a comparison is always tricky because of the insufficient knowledge one researcher has of codes of others. This task is better left to some intercomparison effort where every Author run their code on a common set of benchmarks.

3. **ML**: Finally, my last main remark is that modelling in geoscience is often a compromise between physical accuracy and numerical cost. This paper pays attention to the accuracy and stability of the solution. The numerical cost is considered through the possibility to extend the time step (section 5). However, in a number of surface models, the time step is often constrained by the need to represent the diurnal cycle and the time resolution necessary for the other processes.
Therefore, I would be interested by a discussion about the numerical cost of the different approaches for a fixed time step (typically 10 to 60 minutes for soil or snow models resolving the diurnal cycle). Indeed, the algorithm proposed by the authors requires to solve $k \times m$ linear systems. It means that the improved accuracy (guaranteed convergence and stability) comes with a potentially much more expensive cost than the approaches published by Voller for ice (iterative but without any linear system to solve), or than the DECP approach commonly used in snow models and requiring the solving of only one linear system. In particular, would it be possible to estimate the number of iterations required in the examples provided in the two test cases? How fast is the convergence in the simple case of heat diffusion without any phase change? Is it possible to estimate an average number of iterations in long simulations based on real forcing conditions?

**AR**: The numerics we use is robust, reliable, and realistic (Prentice et al. 2015). It never breaks and it give the required results. In our opinion when using linear methods for the problems under scrutiny, solutions do not match with what is expected for known solutions and mass error are

amplified. But obviously we cannot say for sure of models by other Authors, for the same reasons we have written in the previous answer. For what regards the number of operations performed by the non-linear we did some computation to assess this and our result is reported in the table below. We performed a simulation of 1 year with an hourly time step for the numerical test case reported in Section 5, with different spatial discretizations. The tolerance 10e-11 has been rescaled with the water latent heat of fusion and the water density. The maximum number of iterations for each time step is 40.

| # control volumes | 500 | 1000 | 2000 | 5000 | 10000 |
|---|---|---|---|---|---|
| mean number of iterations NCZ | 12 | 13 | 14 | 16 | 18 |
| mean number of iterations N. R. | 40 | 40 | 40 | 40 | 40 |
| mean number of iterations g. c. N | 40 | 40 | 40 | 40 | 40 |

As you can see the mean number of iterations is higher compared to the results reported in Casulli and Zanolli 2010 and this is because the derivative of the internal energy function is steeper that that the derivative of the water content function that appears in the Richards equation.

We have compared the performance of our code with a simple Newton-Raphson method (N. R., in the second row) and the so called globally convergent Newton (g. c. N., third row). The problem is that they are not faster, simple Newton does not converge so it always reaches the maximum number of iterations allowed, and the solution as presented in Fig. G2 does not reproduce the analytical solution. If it fails to converge, no time saving is obtained.

**Minor comments**

1. **ML**: Lines 49-55: It might also be interesting to mention that the notion of interface is also often meaningless in ice and snow where thick isotherm layers of a mixed solid-liquid medium are very common.

   **AR**: Thank you, we have added it in line 55 of the revised manuscript.

2. **ML**: Line 74: Applications with long time steps also include the surface components of climate models and Numerical Weather Prediction models, and also models dedicated to avalanche hazard forecasting.

   **AR**: Thank you, added in the revised manuscript in line 75.

3. **ML**: Line 115: "SFCC have an inflection point". With just a quick look at Bao et al., 2016, I could not find to which figure or comment this statement refers. Could you provide the details?

   **AR**: From Bao et al 2016, Section 1, paragraph 3 "Currently, the most common problems in frozen soil modeling are unstable simulation and heavy computational cost due to the highly nonlinear relationship in soil temperature, soil moisture, and ice content caused by the substantial latent heat associated with the phase change. Thus, how to deal with this highly nonlinear relationship between these three variables in frozen soil is the key focus of frozen soil model development." They use the term nonlinear relationship.

   We have also added a reference to Hansonn et al. (2004), where Figure 1 shows the apparent heat capacity function with a maximum that correspond to an inflection point in the SFCC function.

4. **ML**: Section 2 is really nice and interesting to read, but a number of symbols definitions come a bit too late after their first use in an equation. For instance, enthalpy already appears in Eq. 1 but is only defined from Eq. 5 to Eq.8. Similarly, latent heat of fusion and liquid water fraction are used in Eq. 3 but are only defined after Eq . 7. It is not critical for the understanding but maybe some reordering could manage to avoid these late definitions.

   **AR**: Eq. 1 – 4 hold in general, independent of the material considered. This part is meant to present the three different formulations and to highlight that Eq.1 best represents the physical system because it expresses the conservation of enthalpy. By contrast, Eq.2 and Eq.3 are derived from Eq.1 by just applying the chain rule of derivatives. The enthalpy for the soil, as well as the liquid water content are presented later since in our opinion it is not necessary to know how the enthalpy function is defined. Proof of this is that the mathematical model, Eq.1,2,3,4, can be applied also for the Neumann problem and the Lunardini one.

5. **ML**: Eq. 9: Although their meaning is relatively obvious, please do not forget to define indexes i and n.

   **AR**: Done

6. **ML**: Line 179: I think Eq. 12 corresponds to the implicit case, not semi-implicit, am I wrong?

   **AR**: No, you are not. There is an error in the apex. It is not n+1 but n and the same applies to Eq. 13.

7. **ML**: Eq. 12: Source terms seem to be expressed at the beginning of the time step in this formalism. In surface models, I think that it is relatively common to express the source terms which depend on temperature at the end of the time step in order to improve the stability especially because of the longwave surface radiation function of surface temperature. I think this does not affect the possibility to apply this algorithm because when these terms are linearized, it just adds terms in the coefficients of the A tridiagonal matrix and in the b vector without changing the formalism. But maybe the authors could just mention that source terms do not necessarily have to be expressed at the beginning of the time step to allow this algorithm to be applied.

   **AR**: As regards the up-welling longwave surface radiation, does it not enter in the governing equation since it represents the boundary condition of the problem (the surface energy budget). In this case it enters in the discretized equation for the uppermost control volume in the numerical flux through the upper boundary representing the soil surface.

   The source/sink term S could be expressed in an implicit manner if the matrix A remains at least positive semidefinite and symmetric matrix. Thus, it is necessary to pay attention that the implicit discretization of S does not affect the feature of A: entries on the main diagonal should be positive, off diagonal should be negative and symmetric. If the requirements on A are fulfilled, then the NCZ algorithm can be used.

8. **ML**: Line 432 It was solved

   **AR**: Done

9. **ML**: Lines 436-438 This is true, but this development has not really been finalized until now and all recent works using Crocus are still based on a simple bucket approach for liquid water percolation.

   **AR**: We have reformulated as "Even though all recent works using Crocus are still based on a simple bucket approach for liquid water percolation (Morin et al., 2012; Lafaysse et al.,2017), D'Amboise et al. (2017) implemented a routine for water flow in the snowpack based on the Richards equation, which is characterized by nonlinear behaviour like the enthalpy equation. To

solve it, they adopted an approach based on Picard iteration with variable time steps (as in Paniconi and Putti, 1994)."

10. **ML**: Table 1 and Appendix A14: I don't really see what is the difference between Crocus and SNOWPACK. I do not understand what means "phase change are accounted for as volumetric heat sinks (melting) and sources (refreezing)" and why a nonlinear solver would be "not required". Is SNOWPACK algorithm really different from the DECP approach? Is it possible to better explain the difference if any?

   **AR**: Also SNOWPACK uses the DECP approach and the manuscript has been changed, accordingly.

11. **ML**: General comment about Appendix A13 and A14: I also think that most multilayer snow schemes simpler than Crocus and SNOWPACK and typically used in climate or hydrology models are also based on this DECP approach. Maybe it could be mentioned somewhere.

   **AR**: We have included in the model review also the snow routine of the ORCHIDEE model (A15) and that of the JSBACH model (A16)

**Typos**

1. **ML**: Line 22 these models,

   **AR**: Thank you, we have corrected it.

2. **ML**: Line 175 orthogonal

   **AR**: Thank you, we have corrected it.

3. **ML**: Line 312 to match the aim

   **AR**: Thank you, we have corrected it.

References

CASULLI, Vincenzo; ZANOLLI, Paola. A nested Newton-type algorithm for finite volume methods solving Richards' equation in mixed form. SIAM Journal on Scientific Computing, 2010, 32.4: 2255-2273.

PRENTICE, I. Colin, et al. Reliable, robust and realistic: the three R's of next-generation land-surface modelling. *Atmospheric Chemistry and Physics*, 2015, 15.10: 5987-6005. https://acp.copernicus.org/articles/15/5987/2015/acp-15-5987-2015.html

HANSSON, Klas, et al. Water flow and heat transport in frozen soil: Numerical solution and freeze–thaw applications. *Vadose Zone Journal*, 2004, 3.2: 693-704.

---

## Author Comment (AC2) · 28 Jan 2021

Dear Anonymous Referee #2,

Thank you very much for your review and your constructive comments. The entire text of your comment is shown (RC) together with our authors' responses (AR).

Kind regards, Niccolò Tubini – on behalf of all authors

**Remarks**

1. **RC**: In the abstract, the authors state that "the nonlinear behaviour of enthalpy as function of temperature can prevent thermal models of snow, ice and frozen soil from converging to the correct solution" but do not provide a description or citation for this claim. Reading further into the paper, this claim is based on a survey of experts, cited as personal communications. I appreciate the point that the authors are trying to make and it is indeed an important advance of the NCZ method, but the phrasing could be improved throughout for clarity.

   **AR**: The description and citation is contained in the introduction and in lines 115-128 of the discussion manuscript. Later, in lines 126-148 we discuss the three forms of the governing equation and the numerical drawbacks that arise when they are solved.

2. **RC**: The main advance that this paper reports is the implementation of a novel algorithm for solving phase change problems. However, the pseudocode included in the paper is not very useful as opposed to a well-documented version of the code that is easy to run. While the code is released publicly, the github documentation is unclear, written in java (for good reason), and not approachable. I was hopeful that I could run the code but that did not seem feasible.

   **AR**: The pseudocode has been rewritten in a clearer way. We also increased the documentation of the code which is provided on Github in the Jupyter_Notebook directory, 00_FreezeThaw.ipynb. Our code must be run inside the OMS3 Console (David et al. 2012) or using the Dockerized version. OMS3 was chosen because it allows to use ancillary components for radiation, calibration and other components. For setting up the environment there are a certain number of steps to be performed which are described on the first Github page.

   The code is written in a clear Java where classes have an extensive name which refers to what the classes do, but we are aware that it could not be so easy to understand the runtime combination in which the classes are used.

   The executable of the version used in the paper are available under Zenodo. For executing them once installed OMS3 please follow the instructions contained in the Jupyter_Notebook /_README.ipynb and Jupyter_Notebook /00_FreezeThaw1D.ipynb for all the details about the simulation input and parameters.

   We added these details also in the revised manuscript in Section Code availability

3. **RC**: This paper neglects to cite or engage with Schoof and Hewitt (2016), who derive a general enthalpy model for phase change. Schoof and Hewitt (2016) follows on from Aschwanden and Blatter (2009) and Aschwanden et al. (2012), where only the first paper is referenced in the manuscript. Beyond the numerical implementation of the NCZ method, it is not clear what this paper adds beyond Schoof and Hewitt (2016) in terms of physical understanding and the role of enthalpy in phase change.

**AR**: We inserted citations to the above papers and also Hewitt and Schoof 2017, where the authors present the numerics of their algorithm for the first time. The mathematical description of glaciers using enthalpy has enthalpy in common with our permafrost model, but the other aspects of the physical problem differ. For example, heating is generated by internal friction of the flowing ice. We have only external forcing driving our physics and, at present, no fluid flow. What we improved with respect to the shared challenge of enthalpy is the treatment of its nonlinear dependencies on temperature. Specifically, the derivative of enthalpy is non monotonic and cannot be integrated by a traditional Newton algorithm and a globally convergent Newton method has to be used instead. Many of these methods were implemented in the past by using "tricks" (Paniconi et al., 1994, Dall'Amico et al. 2011,) because a safely convergent method was unavailable. The more evident result is that we do not need to track where the front between ice and water (cold and temperate ice) is. With NCZ, we do not need a section like the Section 3 in Schoof and Hewitt 2016, which is required to preserve properly the mass in absence of an appropriately convergent algorithm, and we do not need to use an algorithm to solve the energy equation like that one presented in Appendix A.

**Specific comments**

1. **RC**: I suggest replacing the title with: "An undated numerical method for solving the heat equation with phase change".

   **AR**: We thank the reviewer, but we prefer to stick with our original title. It is long but, in our opinion, clearer in conveying the usefulness of out method.

2. **RC**: line 105: is theta_s defined anywhere?

   **AR**: We have added the definition of theta_s

3. **RC**: line 115: the kink in the sfcc only matters if the authors take the derivative, which is not required! I am not sure about the value of this `straw-man' argument about the three 'identical yet different' representations of the heat equation. First off, the language is unclear, so it is opaque as to what method the authors will actually use.

   **AR**: To solve a nonlinear system it is necessary to use the derivative to linearize the function, as required by Newton type algorithms. This is a general comment meant to say that the treatment of this nonlinear relationship between temperature and enthalpy is challenging independently of the numerical scheme (finite differences, finite volumes or finite element) one would adopt. The equation form we are going to solve is stated at lines 69-73 of the submitted manuscript.

4. **RC**: line 163: semi-implicit is not required, implicit is required. semi-implicit is a convenient method of mixing explicit and implicit methods to decrease time step restrictions.

   **AR**: The thermal conductivity is a function of the solution and therefore, by using a full implicit discretization one obtains a fully nonlinear system of equation. To solve it, it is possible to use a

Picard iteration (Casulli and Zanolli, 2010). In this case we adopted only one Picard iteration obtaining a semi-implicit discretization, but yes, this is just a legitimate option and we added in the Appendix C of the revised manuscript.

5. RC: line 169: ok, let me get this straight: the authors asked their colleagues if there is guaranteed convergence for nonlinear problems using the `currently used algorithms' and they said no? Tell me more. Tell me why convergence is not guaranteed and how NCZ guarantees it - don't refer me to their paper. That is not the point. All the authors need to say is that NCZ offers advantages. Instead the authors generate an entire table showing that all of the methods they can think of have drawbacks, based on the word of their colleagues? The articulation of this argument needs substantial bolstering.

AR: The convergence, assured by the NCZ algorithm, is only fully realized with a numerical scheme formulated to be conservative. Given the number of established models representing temperature and phase change, and how central the issue is for cryosphere research, some readers may be left with the impression that there must be a model that is conservative and guaranteed to converge. Appendix A and its summary in Table 1 shows that this is not the case based on: (1) the form of the initial equations and (2), when available, statements by the authors of the algorithms. Actually, we cannot determine here where the theoretical limitations might translate into relevant practical consequences, and, besides we show that the problem we solve is shared across differing fields of cryospheric modelling.

In order to be clearer, we have updated the table adding the references to papers and added a reference to Appendix A in the caption. For clarity, we will also change line 148 (discussion manuscript) to "A summary of relevant models is given in Table 1 and more details in Appendix A".

As we discuss more deeply in the revised manuscript, the rational of (1) is discussed in Casulli and Zanolli, 2010; Nicolski et al., 2007; Voller, 1990. Moreover, the work by Roe states that the only way to preserve the chain rule at the discrete level is to respect Eq. 9 (of our manuscript) that leads to solve the so-called enthalpy form of the equation. Richards' equation presents the same numerical issues, convergence problems arise when solving the so-called psi-based equation, i.e., the form in which the time derivative is expressed in the form of

$$\frac{\partial \theta}{\partial t} = \frac{\partial \theta}{\partial \psi} \frac{\partial \psi}{\partial t}$$

This shortcoming it is not only stated in Casulli and Zanolli 2010, but also in D'Amboise et al. 2017 where they solve the Richards equation to simulate water flow in the snowpack.

About the DECP approach, Nicolsky et al. 2007, Section 2.3 "One of the consequences of this two-step procedure is that the region where the phase change occurs can be artificially stretched, leading to inaccuracies in the simulation of active layer depth".

6. **RC**: line 195: it looks like it comes down to the fact that the enthalpy is, for some reason, not monotonic with temperature, but isn't that the reason to use the enthalpy: because it is monotonic?

I agree that at the melting temperature there is a jump in enthalpy governed by the latent heat, but does that mean that it is not monotonic?

**AR**: The enthalpy function is monotonic sure, but what we need is that its derivative be monotonic within any neighbourhood of the root for any Newton type algorithm to converge. This in principle is not a problem if the initial guess is carefully chosen to be sufficiently close the solution and, in an interval where the derivative is monotonic. However, the drawback of an unhappy choice of the initial guess is shown in Figure G1: the algorithm enters in a loop without reaching the convergence. Therefore, the so-called globally convergent Newton methods rely on a trick that consists in reducing with a damping factor the increment to calculate the new approximation, but this search is not guaranteed to succeed in advance and in the favorable situations implies many trial and search. The advantage on the NCZ algorithm is that it guarantees convergence of the solution for any choice of the initial guess. Figure G2 shows the numerical solution obtained with the Newton algorithm, the Newton algorithm with a dampening factor and the NCZ. Figure G3 shows that the choice of the dampening factor affects the numerical solution, and it is a source of uncertainty.

We tried to make it clear in the text as we added at the beginning of section 3.2 "Difficulties in solving the nonlinear system of Eq. (16) arise from the non-monotonic behaviour of the derivative of the enthalpy, h(T), with respect to temperature, and because for some parametrizations used for substances - like water - the derivative of the enthalpy is not correctly defined."

7. **RC**: line 202: I must be very confused, why don't you just solve for the enthalpy and use the jump conditions to determine the temperature (Schoof and Hewitt, 2016, SH2016)?

**AR**: When following the approach of SH2016, we need to capture the moving boundary separating the two domains (liquid water and ice), an operation that is computationally expensive and difficult to implement, as stated in lines 44-69 of our submitted manuscript. Another reason is that in the SH2016 algorithm: (a) temperature cannot be larger than $T_m$. This is not acceptable in models where soil and soil moisture can experience temperatures larger than $T_m$. This means that Eq. (19) cannot be used since in case of $h^{n-1} \geq \rho c(T_m - T_{ref})$, $T$ cannot be assigned a priori. (b) T is defined accordingly to the value of h at the previous time step (Hewitt and Schoof, 2017, HS2017 Eq. 19)

Moreover, in HS2017 Eq. 19 Section 3 "All the terms in $\nabla \cdot Q$ are discretized explicitly", a procedure which causes restriction in choosing the time step which must be controlled.

Furthermore, we like to point out that in SH2016 and HS2017, the numerical test is performed for a steady state problem, where the time derivative is 0. Our cases are non-stationary and the NCZ algorithm is used because of the nonlinear behaviour of the terms that comes from the discretization of the time derivative. These terms in a steady state problem do not exist.

8. **RC**: section 4: I have no idea what the Neumann and Lunardini solutions are: describe the problems physically? I can certainly look in the appendix (and did) to find the mathematics, but until I saw Figure 2, I was totally confused at what problem you were trying to solve.

**AR**: We added a description of the problem in the text for both the problems.

9. **RC**: line 288: SUTRA uses an $\varepsilon$ in the enthalpy function as well?

**AR**: As reported in Kurylyk, et al., 2014, Section 3

"*Hence, SUTRA and other cold region thermohydraulic models generally utilize some form of a soil freezing curve that considers freezing over a range of temperatures less than 0°C. However, the previously detailed analytical solutions employ the crude assumption that the soil freezing curve is represented as a step function. It is difficult to employ a step function soil freezing curve in a numerical model because the apparent heat capacity in the zone of freezing or thawing is dependent on the slope of the soil freezing curve[4,5], which would be infinite for a step function. A very steep piecewise linear soil freezing curve was employed in SUTRA to approximate a saturated step function soil freezing curve.*"

10. **RC**: table 2: there does not seem to be monotonic convergence. given that this paper is claiming guaranteed convergence, I would have liked to see a convergence plot showing that the solution does converge at the power of the discretization, both in space and time. Also, it is worth mentioning the error order for both, especially since the method is first-order in time! predictor-corrector methods (or Heun's method) could be used instead of Crank-Nicholson to increase the resolution without the same time step restrictions.

**AR**: Actually, Table 2 is not referring to the convergence of the algorithm. Instead, the convergence rate is for the errors in the NCZ algorithm. Moreover, the solution of the zero-isotherm is obtained by interpolation of the numerical solution. To make it clearer, we added to the text: "For the numerical solution the position of the thawing front has been reconstructed from the linear interpolation of the temperature profile. Table 2 reports the deviations of the reconstructed position of the zero-isotherm from the analytical solution."

11. **RC**: figure 4: if the point is to show that the left and right panels are the same, then I suggest, plotting them on one panel and using the other panel to show the difference.

**AR**: In the resubmitted manuscript we now show the difference between the numerical and analytical solutions.

12. **RC**: section 4.2: what defines the mushy zone in the Lunardini analytical solution? and how is this different than Katz (2008)?

**AR**: Oeterling and Watts (2004, in Katz, 2008) discuss the mushy region referring to the development of the ice sheet. The mushy zone is characterized by an increase of solutes concentration, primarily salt but also anthropogenic pollutants, with a consequent variation of the density. This gradient density '*provide the potential energy to drive convection within the interstices of the ice matrix and the water below the ice*'.

Referring to the Lunardini analytical solution, the mushy zone is used to indicate the transition zone between where ice and liquid water coexist in varying proportions in the soil. In the Lunardini problem neither the water flow nor solutes concentration are considered. Thus, the variation of

water density due to the expulsion of solutes, and the consequent convection flow is not considered. To avoid possible misunderstanding, we changed 'mushy zone' to 'partially frozen zone'.

13. **RC**: line 329: is this a paragraph fragment?

    **AR**: We have corrected it.

14. **RC**: Most figures: the axis labels as well as figure text are missing letters and difficult to read.

    **AR**: Sorry for this inconvenient. We have corrected the Figures.

15. **RC**: line 510: why $\varepsilon$ is required? It seems that the value of the enthalpy is that there is a smooth transition across the phase change - adding $\varepsilon$ negates the authors' claim that they are `guaranteeing energy conservation', because they have added a fictious mushy zone.

    **AR**: There is no way to avoid the introduction of $\varepsilon$ since the enthalpy function needs to be continuously differentiable and enthalpy function with a step change at the melting temperature is not. See assumption C1 on the apparent heat capacity function, lines 204-206 of the submitted manuscript.

    However, the temperature range $\varepsilon$ can be chosen sufficiently small in order to make this approximation negligible when compared to the physical behaviour of water, considering that: (a) The melting of water in temperate ice is known to actually occur progressively below 0ºC along grain boundaries (Langham 1974; Nye and Frank 1973). (b) Freezing often occurs below the melting point. (c) In porous media such as soil, ice melts across a range of temperatures due to the Gibbs-Thompson effect in pores and surface affects at the interfaces between ice and particles (Rempel et al., 2004; Watanabe and Mizoguchi 2002).

References

Akyurt, M., Zaki, G., & Habeebullah, B. (2002). Freezing phenomena in ice–water systems. Energy conversion and management, 43(14), 1773-1789.

Dall'Amico, M., Endrizzi, S., Gruber, S., & Rigon, R. J. T. C. (2011). A robust and energy-conserving model of freezing variably-saturated soil. *The Cryosphere*, *5*(2), 469-484.

Kurylyk, B. L., McKenzie, J. M., MacQuarrie, K. T., and Voss, C. I.: Analytical solutions for benchmarking cold regions subsurface water flow and energy transport models: One-dimensional soil thaw with conduction and advection, Advances in Water Resources, 70, 172–184, 2014

Langham, E. J. 1974. "Phase Equilibria of Veins in Polycrystalline Ice." Canadian Journal of Earth Sciences 11 (9): 1280–87.

Nicolsky, D., Romanovsky, V., Alexeev, V., and Lawrence, D.: Improved modeling of permafrost dynamics in a GCM land-surface scheme, Geophysical Research Letters, 34, 2007, Section 2.3

Nye, JF, and FC Frank. 1973. "Hydrology of the Intergranular Veins in a Temperate Glacier." Symposium on the Hydrology of Glaciers 0 (1): 157–61.

Oertling, A. B., and R. G. Watts (2004), Growth of and brine drainage from NaCl-H2O freezing: A simulation of young seaice, J. Geophys. Res.,109, C04013, doi:10.1029/2001JC001109.

Paniconi, C., & Putti, M. (1994). A comparison of Picard and Newton iteration in the numerical solution of multidimensional variably saturated flow problems. Water Resources Research, 30(12), 3357-3374.

Rempel, Alan W., J. S. Wettlaufer, and M. Grae Worster. 2004. "Premelting Dynamics in a Continuum Model of Frost Heave." *Journal of Fluid Mechanics* 498 (January): 227–44. https://doi.org/10.1017/S0022112003006761.

Watanabe, Kunio, and Masaru Mizoguchi. 2002. "Amount of Unfrozen Water in Frozen Porous Media Saturated with Solution." *Cold Regions Science and Technology* 34 (2): 103–10. https://doi.org/10.1016/S0165-232X(01)00063-5.

Wilson, P. W., Heneghan, A. F., & Haymet, A. D. J. (2003). Ice nucleation in nature: supercooling point (SCP) measurements and the role of heterogeneous nucleation. Cryobiology, 46(1), 88-98.

---

## Author Comment (AC3) · 28 Jan 2021

Dear Dr. Matthieu Lafaysse,

Thank you for your clarification.

Kind regards, Niccolò Tubini – on behalf of all authors

---

## Author Response (AR2)

Dear Editor,

We appreciate the clear language of Anonymous Referee #2 and the opportunity to further clarify and revise. Our intention is certainly not to "bash". Rather, the effort to include Table 1 and Appendix G has been deliberate. It serves to illustrate the relevance of what we present and our careful consideration of existing approaches.

After describing advantages and challenges of differing formulation and solvers, we use this summary of prominent models to demonstrate how our work relates to current simulation practice. A reader without detailed knowledge about solving heat transfer with phase change may otherwise suspect an established way of reliably finding conservative solutions to be 'somewhere out there', given the many established models that exist, and the fundamental nature of the problem.

We have now further clarified in the Table caption that identifying theoretical limitations does not imply the models we list are not fit for application. We hope that will satisfy any remaining concerns.

Kind regards,
Niccolò Tubini, on behalf of all authors

**List of relevant changes**

- Line 73 we changed 'a semi-implicit' with 'an implicit'.
- Line 149 we reformulated the sentence.
- Line 166 we changed 'a semi-implicit' with 'an implicit'.
- Line 168 we added the reference Casulli and Zanolli 2012.
- Lines 171-172 we reformulated the sentence in order to clarify the expression 'guaranteed convergence'.
- Table 1 we reformulated the caption.
- Table 1 we changed the header of the fifth column from 'Limitation' to 'Theoretical limitation'.
- Table 1 we changed the expression 'non-convergence problem' with 'non-convergence'.
- Table 1 we changed the expression 'Convergence is not guaranteed' with 'Nonlinear solver'.
- Table 1 we change the footnote (g).
- Line 185 we changed 'a semi-implicit' with 'an implicit'.
- Equation (12, 13) we changed $\Lambda_j^n$ with $\Lambda_j^{n+1}$
- Line 426 we reformulated the sentence.
- Line 436 we reformulated the sentence.
- Line 464 we reformulated the sentence.
- Line 480 we reformulated the sentence.
- Line 486 we reformulated the sentence.
- Appendix C we changed the name.
- Table E1 we removed the footnote and corrected the initial and the surface temperature values.